# Marine-Derived Bisindoles for Potent Selective Cancer Drug Discovery and Development

**DOI:** 10.3390/molecules29050933

**Published:** 2024-02-21

**Authors:** Mengwei Xu, Zhaofang Bai, Baocheng Xie, Rui Peng, Ziwei Du, Yan Liu, Guangshuai Zhang, Si Yan, Xiaohe Xiao, Shuanglin Qin

**Affiliations:** 1Hubei Engineering Research Center of Traditional Chinese Medicine of South Hubei Province, School of Pharmacy, Xianning Medical College, Hubei University of Science and Technology, Xianning 437100, China; x690385686@163.com (M.X.); qipp0207@163.com (R.P.); duziwei0101@163.com (Z.D.); liuyan20210915@163.com (Y.L.); zhanggs1999@163.com (G.Z.); yansi92674@163.com (S.Y.); 2Senior Department of Hepatology, The Fifth Medical Center of PLA General Hospital, Beijing 100039, China; baizf2008@hotmail.com; 3Department of Pharmacy, The Tenth Affiliated Hospital of Southern Medical University (Dongguan People’s Hospital), Dongguan 523059, China; baochengxie@126.com; 4China Military Institute of Chinese Materia, The Fifth Medical Center of PLA General Hospital, Beijing 100039, China

**Keywords:** bisindole alkaloids, marine natural products, cancer, antitumor activity, anticancer drug

## Abstract

Marine-derived bisindoles exhibit structural diversity and exert anti-cancer influence through multiple mechanisms. Comprehensive research has shown that the development success rate of drugs derived from marine natural products is four times higher than that of other natural derivatives. Currently, there are 20 marine-derived drugs used in clinical practice, with 11 of them demonstrating anti-tumor effects. This article provides a thorough review of recent advancements in anti-tumor exploration involving 167 natural marine bisindole products and their derivatives. Not only has enzastaurin entered clinical practice, but there is also a successfully marketed marine-derived bisindole compound called midostaurin that is used for the treatment of acute myeloid leukemia. In summary, investigations into the biological activity and clinical progress of marine-derived bisindoles have revealed their remarkable selectivity, minimal toxicity, and efficacy against various cancer cells. Consequently, they exhibit immense potential in the field of anti-tumor drug development, especially in the field of anti-tumor drug resistance. In the future, these compounds may serve as promising leads in the discovery and development of novel cancer therapeutics.

## 1. Introduction

Cancer is one of the most significant causes of death worldwide, affecting various diseases in any part of the body. A key characteristic of cancer is the rapid production of abnormal cells that grow beyond their normal boundaries, potentially invading adjacent body parts and spreading to other organs. This spreading, known as metastasis, is the primary cause of cancer-related deaths [1]. Despite advances in early detection, surgical techniques, and targeted therapy, challenges in cancer treatment persist, including low efficacy, high toxicity, and drug resistance, severely impacting patients’ quality of life. Consequently, the search for efficient, safe, and highly selective anti-cancer compounds has become a crucial aspect of contemporary cancer research [2].

The sea covers over 70% of the Earth’s surface and is a land rich in resources. The marine ecosystem, which hosts a large number of organisms, is one of the regions with the highest biodiversity on earth [3]. For a long time, countries have invested substantial human, material, and financial resources in developing and utilizing marine resources. The immense diversity of marine organisms has proven to be a rich source of inspiration for drug discovery. The success rate of marine natural product derivatives is four times higher than that of other natural derivatives [4]. The biodiversity of marine organisms provides abundant resources for discovering and developing new anticancer drugs for human malignant tumors [5]. To date, 20 drugs derived from marine sources are used clinically, of which 11 exhibit antitumor effects [6].

Indole is a multifunctional active pharmacophore and a heterocyclic compound widely present in natural and synthetic compounds with biological activity. Indole alkaloids from natural sources display diverse mechanisms and structures and exert anticancer potential through various antiproliferation mechanisms. Thus, indole alkaloids play a significant role in the discovery of new anticancer drugs [7,8,9,10,11]. Scientists have subsequently isolated various new bisindole alkaloids from marine organisms, especially deep-water sponges. They are usually extracted with organic solvents (e.g., methanol, ethyl acetate). And the extracts were concentrated and partitioned between organic and aqueous phases. The organic phase is separated with chromatography separation technology, including silica gel column chromatography, high-performance liquid chromatography, etc. Most marine-derived bisindoles exist in solid form, which makes it convenient for us to determine their absolute configuration by single crystal. A number of marine-derived bisindoles exhibit strong and varied biological activities. Due to their unique biological activities and chemical structures, they have become a research focal point in pharmaceutical chemistry as lead compounds for new drug development. It is noteworthy that many drugs based on marine-derived bisindoles have been approved or are currently in clinical research, such as midostaurin, lestaurtinib, and enzastaurin. These drug molecules have shown potent selective anti-tumor effects. This paper reviews the recent research progress on marine bisindole alkaloids and their derivatives with anticancer activity.

## 2. Topsentin Family Bisindole Alkaloids and Their Derivatives

Topsentins family bisindole alkaloids are compounds that utilize carbonyl imidazole as a spacer to link two indole moieties, predominantly isolated from deep-sea sponges.

A research team isolated topsentin B_1_ (**1**) (Figure 1) from a sponge, which exhibited significant cytotoxicity to the P388 cancer cell line with an IC_50_ of 4.1 ± 1.4 µM. It also demonstrated a moderate inhibitory effect on human HL-60 cancer cells, with an IC_50_ of 15.7 ± 4.3 µM, and exhibited anti-proliferative effects on human tumor cells HCT-8, A-549, and T47D. In vivo tumor models established in mice confirmed that topsentin B_1_ had an inhibitory effect on P388 and B16. Further research on P388 cancer cells revealed that the compound could inhibit DNA (deoxyribonucleic acid) (91%) and RNA (ribonucleic acid) synthesis (57%), while not affecting protein synthesis [12,13].

Topsentin B_1_ (**1**) and topsentin B_2_ (**2**) (Figure 1) exhibited moderate anti-proliferative activity against NSCLC-N6 (non-small cell lung cancer), with IC_50_ values of 12 and 6.3 µg/mL, respectively. They also displayed cytotoxicity to HeLa cells, with IC_50_ values of 4.4 and 1.7 µM, respectively [14,15,16]. (Topsentin B_2_, also known as bromotopsentin).

The four bisindole alkaloids topsentin B_2_ (**2**), 4,5-dihydro-6″-deoxybromotopsentin (**3**), deoxytopsentin (**4**), and bromodeoxytopsentin (**5**) (Figure 1), extracted from the sponge Spongosorites sp., all showed moderate activity (IC_50_: 1.1–7.4 µg/mL) against AGS (adenocarcinoma) and L1210 (lymphocytic leukemia). Compounds **4** and **3** demonstrated moderate anti-proliferative activity on BC (breast cancer) cancer cells, with IC_50_ values of 10.7 and 17.5 µg/mL, respectively. Compound **4** also exhibited moderate activity against HepG2, with an IC_50_ of 3.3 µg/mL [17]. It was previously reported that topsentin B_2_ had cytotoxicity to P388 cells, with an IC_50_ of 7.0 µg/mL [18].

Bao’s research team isolated bromodeoxytopsentin (**5**) and isobromodeoxytopsentin (**6**) (Figure 1) from marine sponge sp., which were cytotoxic to K562 cells with IC_50_ values of 0.6 and 2.1 µg/mL, respectively [19]. Compound **6** is cytotoxic to five human cancer cell lines: A549, SK-OV-3, SK-MEL-2, XF498, and HCT15 [20].

In 1988, while extracting natural products from topsentins, several analogues were synthesized. Under identical conditions, the cytotoxicity of natural products **1**–**4** and analogues **7**–**10** (Figure 1) was evaluated. The IC_50_ values for natural products **1**–**4** against P388 cancer cells were 2.0, 12.0, 4.0, and 7.0 µg/mL, respectively, while those for analogues **7**–**10** were 4.0, 0.3, 2.5, and 1.8 µg/mL, respectively. The results indicated that the introduction of a hydroxyl group enhanced cytotoxicity, whereas the introduction of a bromine substituent reduced it [12].

Azaindole and its derivatives have demonstrated significant biological activities in previous studies, including antitumor effects [21,22,23,24,25]. A team reported synthesizing a series of new 1,2,4-oxadiazole topsentin analogues and screened five compounds, **11**–**15** (Figure 1), which showed strong activity on pancreatic ductal adenocarcinoma cell lines (including Panc-1, Capan-1, and SUIT-2), with EC_50_ values ranging from 0.4 to 6.8 µM. None of these derivatives exhibited cytotoxic effects against non-tumor pancreatic cells. Further studies revealed that the anti-proliferation mechanism of compounds **11**–**15** involved promoting apoptosis, which was related to the externalization of phosphatidylserine in the cell membrane, suggesting that their target was GSK3β (glycogen synthase kinase 3 beta) [26].

The existence of a carbonyl spacer group in the topsentin family allows small-molecule compounds to have greater flexibility, better adapt to the ATP (adenosine triphosphate) binding site of CDK1 (cyclin-dependent kinase 1), and act as a hydrogen bond receptor. This enables interaction with amino acid residues at the active site of CDK1. The designed and synthesized derivative **16** (Figure 1) exhibited a significant anti-proliferative effect against PDAC (pancreatic ductal adenocarcinoma cells) (Hs766T, HPAF-II, PDAC3, and PATU-T), with IC_50_ values of 5.7 ± 0.60, 6.9 ± 0.25, 9.8 ± 0.70, and 10.7 ± 0.16 µM, respectively. This compound promoted apoptosis in PDAC cells and regulated the expression of CDK1. ADME (absorption, distribution, metabolism, and excretion) prediction indicated good pharmacokinetic properties, warranting further clinical investigation [27].

## 3. Nortopsentin Family Bisindole Alkaloids and Their Derivatives

Nortopsentins A–C (**17**–**19**) (Figure 2) possess a 2,4-bis(3′-indolyl)imidazole structure skeleton, which exhibits significant inhibitory effects against P338 cells, with IC_50_ values of 7.6, 7.8, and 1.7 µg/mL, respectively [18]. In recent years, the unique structure and anticancer potential of nortopsentins bisindole alkaloids have garnered the attention of many research teams, leading to a series of modifications. The modifications primarily target the middle part (five-membered heterocycle) of the two indole structures and further extend to one or two indole units [28].

Patrizia Diana’s team reported the synthesis of bisindole thiophene and bisindole pyrazole derivatives. Several of these compounds were selected for evaluation against 60 cancer cell lines from nine human cancer types. The results indicated that the most active bisindole thiophene compound was **20** (Figure 2), particularly effective against leukemic cell lines (CCRF-CEM, MOLT-4, HL-60, RPMI-8226, K-562) with GI_50_ values ranging from 0.34 to 3.54 µM. It demonstrated good selectivity for the HT29 and HCC-2998 cell lines of the colon cancer subgroup, with GI_50_ values of 2.79 and 2.83 µM, respectively. It also showed selectivity for NCI-H522 of the non-small cell lung cancer subgroup, LOXIMVI of the melanoma subgroup, and UO31 of the renal cancer subgroup [29]. Bisindole pyrazole **21** (Figure 2) exhibited antitumor activity on most human cell lines, while compound **22** (Figure 2), containing more chlorine groups than compound **21**, was effective on all tested cells [30]. The team also synthesized a series of 2,5-bis(3′-indolyl)furan and 3,5-bis(3′-indolyl)isoxazole antitumor drugs. The in vitro antitumor activities of these compounds against 10 human tumor cells were tested, revealing that all compounds exhibited antiproliferative activity at the highest test concentration of 100 µg/mL. Further tests on compounds **23** and **24** (Figure 2), which showed strong activities in 29 cell lines, revealed that compound **24** exhibited a high level of tumor selectivity, and compound **23** demonstrated selective activity against A549, LXFA629L, and UXF1138L [31].

A series of compounds were synthesized by replacing the imidazole ring with a pyrrole ring. All compounds exhibited anti-proliferative activity at the highest test concentration of 100 µg/mL. Among them, compounds **25** and **26** (Figure 2) demonstrated the most significant anti-proliferative activity against a group of human tumor cell lines (42 human tumor cells) cultured in vitro, with an average IC_50_ value of 0.37 µg/mL. Moreover, in vitro clonogenic assays revealed significant tumor selectivity for both compounds, particularly for compound **26** [32].

New 2,5-bisindolyl-1,3,4-oxadiazoles were designed and synthesized. Compounds **27** and **29** (Figure 2) exhibited strong anticancer activity against the MCF-7 cell line, with IC_50_ values of 1.8 ± 0.9 and 2.6 ± 0.89 µM, respectively. Compounds **27**, **28**, and **29** (Figure 2) displayed potent cytotoxicity against HeLa cells, with IC_50_ values of 9.23 ± 0.58, 9.4 ± 0.37, and 6.34 ± 0.56 µM, respectively. Notably, compound **29** demonstrated significant cytotoxicity against the lung cancer cell line A549, with an IC_50_ value of 3.3 ± 0.85 µM. Importantly, these compounds showed no cytotoxicity against normal human embryonic kidney cells, HEK-293 [33].

A series of bisindole thiadiazole compounds was synthesized and tested for in vitro cytotoxicity against six human cancer cell lines: prostate (PC3, DU145, and LNCAP), breast (MCF-7 and MDA-MB-231), and pancreas (PACA2). The results indicated that substituents at the N-1 and C-6 positions of the indole ring were crucial for inducing cytotoxicity and selectivity towards specific cancer cell lines. Among them, compound **30** (Figure 2), with a 4-chlorobenzyl group and a 5-methoxy substituent, showed significant inhibitory effects on all cancer cell lines tested (IC_50_ range: 14.6–369.8 µM), particularly on LNCAP cancer cell lines (IC_50_ = 14.6 µM) [34].

With further research, a series of new thiazole nortopsentin analogues were synthesized. To increase water solubility, the nitrogen atom of the indole, or 7-azaindole, was modified with a 2-methoxyethyl chain. Among these, four derivatives (**31**–**34**) (Figure 2) demonstrated potent anti-proliferative activity against nearly all 60 human tumor cell lines (GI_50_ range: 0.03–98.0 µM). Further studies revealed that the anti-proliferation mechanism of these compounds against MCF-7 cells involved promoting apoptosis, associated with the externalization of phosphatidylserine in the cell membrane and DNA fragmentation. Compound **31**, exhibiting the strongest activity and selectivity, constrained living cells to the G2/M phase and significantly inhibited the activity of CDK1 in vitro [35].

Compounds **35** and **36** (Figure 2) displayed anti-proliferative activity against about 60 human tumor cell lines, with GI_50_ values of 0.03–13.0 µM and 0.04–14.2 µM, respectively, and did not significantly affect the survival rate of normal human liver cells, indicating high selectivity towards tumor cells. After treating HepG2 cells with these compounds, cell cycle distribution studies showed concentration-dependent accumulation in the sub-G0/G1 phase, restricting living cells to the G2/M phase. The anti-proliferation mechanism involved promoting apoptosis, linked to the externalization of phosphatidylserine in the cell membrane and mitochondrial dysfunction [36].

Compounds **37**, **38**, and **39** (Figure 2) significantly inhibited the activity of CDK1, with IC_50_ values of 0.89 ± 0.07, 0.75 ± 0.03, and 0.86 ± 0.04 µM, respectively, and did not substantially interfere with the proliferation of human normal fibroblasts. In a nude mouse tumor model, these three compounds were also found to have significant inhibitory effects on tumor cell proliferation and growth. Further investigation into their mechanism of action revealed that they could block the cell cycle in the G2/M phase by inhibiting CDK1 activity, increasing the rate of apoptosis, and reducing the phosphorylated form of the anti-apoptotic protein survivin. Additionally, compound **38** synergized with paclitaxel, significantly inhibiting the survival of DMPM (diffuse malignant peritoneal mesothelioma) cells [37].

A series of seven newly synthesized compounds, **40**–**46** (Figure 2), inhibited the growth of four cancer cell lines: MDA-MB-231, MIAPACA-2, PC3, and STO. Among these, compound **45** exhibited the highest activity against malignant peritoneal mesothelioma (STO) cell lines. Further investigation into the mechanism of compound **45** revealed that its treatment of STO cells significantly reduced CDK1 activity in a time-dependent manner and could induce cell cycle arrest in the G2/M phase, accompanied by marked caspase-dependent apoptosis. The phosphorylated form of the anti-apoptotic protein survivin also showed a time-dependent decrease. Additionally, compound **45** synergized with paclitaxel due to the enhanced induction of caspase-3-mediated apoptosis [38].

A new nortopsentin analog was synthesized by partially replacing the central imidazole ring with 1,2,4-oxadiazole and an indole moiety with 7-azaindole. Among the synthesized derivatives, compounds **47** and **48** (Figure 2) exhibited significant cytotoxicity against MCF-7, HCT-116, and HeLa cells (IC_50_: 0.65–13.96 µM). Further studies on the anti-proliferation mechanism in MCF-7 cells showed that they could promote apoptosis, which was associated with the externalization of phosphatidylserine in the cell membrane, chromatin condensation, and membrane blebbing. They also induced cell accumulation in the G0-G1 phase [39].

A team synthesized a new thiazole nortopsentin analog in which the naphthyl group partially replaced the original indole part and the other indole was replaced or retained by 7-azaindole. Three derivatives, **49**–**51** (Figure 2), demonstrated effective anti-proliferative activity against MCF-7 cells, with IC_50_ values of 2.13 ± 0.12, 3.26 ± 0.19, and 5.14 ± 0.34 µM, respectively. Further cytotoxicity studies on MCF-7 cells showed that all three compounds promoted apoptosis, inducing cells to transition towards early apoptosis without necrosis. They also caused cell cycle disruption, significantly reducing the percentage of G0/G1 and S phase cells, increasing the percentage of G2/M phase cells, and leading to the appearance of a sub-G1 cell population [40].

## 4. Hamacanthin Family Bisindole Alkaloids and Their Derivatives

Bao’s research team isolated the hamacanthin family of bisindole alkaloids from marine sponges sp. Compounds such as (3S,6R)-6′-debromo-3,4-dihydrohamacanthin A (**52**), (3S,6R)-6″-debromo-3,4-dihydrohamacanthin A (**53**), trans-3,4-dihydrohamacnathin A (**54**), (S)-hamacanthin A (**56**), (R)-6″-debromohamacanthin A (**57**), (S)-6′,6″-didebromohamacanthin A (**59**), (R)-6′-debromohamacanthin B (**60**), (R)-6″-debromohamacanthin B (**61**), (R)-6′,6″-didebromohamacanthin B (**62**), (S)-hamacanthin B (**63**), (3S,5R)-6″-debromo-3,4-dihydrohamacanthin B (**64**), (3S,5R)-6′-debromo-3,4- dihydrohamacanthin B (**65**), cis-3,4-dihydrohamacanthin B (**66**) (Figure 3) is cytotoxic to five human cancer cell lines: A549, SK-OV-3, SK-MEL-2, XF498, and HCT15. (R)-6′-debromohamacanthin A (**58**) (Figure 3) exhibited a weak inhibitory effect against HCT15 [19,20,41]. A study has shown that compound **58** can inhibit VEGF (vascular endothelial growth factor)-induced expression of MAPKs (p38, ERK, and SAPK/JNK) and the PI3K/AKT/mTOR signaling pathway [42]. (Note: the naming of compounds **57** and **58** here is subject to reference [20]).

The bisindole alkaloids trans-4,5-dihydrohamacanthin A (**55**), hamacanthin A (**56**), **57**, 6″-debromohamacanthin B (**61**), and hamacanthin B (**63**) were extracted from the sponge Spongosorites sp. These compounds demonstrated moderate activity against AGS and L1210 (IC_50_: 1.1–9.0 µg/mL) [17].

The pyrazinone bisindole compound **67** (Figure 3) synthesized by Jiang’s team demonstrated a strong inhibitory effect on a variety of tumor cell lines, including leukemia, non-small cell lung cancer, colon cancer, CNS (central nervous system) cancer, ovarian cancer, kidney cancer, and breast cancer cell lines, with GI_50_ values ranging from 6.6–74.8 µM [43].

## 5. Dragmacidin Family Bisindole Alkaloids and Their Derivatives

The IC_50_ value of dragmacidin (**68**) (Figure 4) extracted from deep-sea sponge against P388 cells is 15 µg/mL, and against A-549, HCT-8, and MDAMB cells is 1–10 µg/mL [44].

Dragmacidon A (**69**) (Figure 4) was isolated from the Pacific sponge *Hexadella* sp. In vitro cytotoxicity tests revealed that **69** exhibited cytotoxicity in the L1210 assay (ED_50_ was 10 mg/mL) [45]. Dragmacidon A has significant anti-cancer activity against A549, HT29, and MDA-MB-231, with IC_50_ values of 3.1, 3.3, and 3.8 µM, respectively. Research has shown that it can inhibit the mitosis of tumor cells by inhibiting PP1 and/or PP2A phosphatase. This may represent a new paradigm to inhibit Ser-Thr PPs type 1 and possibly other analogous phosphatases such as PP2A [46].

Dragmacidin G (**70**) (Figure 4), composed of a pyrazine ring and two indole rings, was isolated from sponges of the genus Spongosorites. In tests using PANC-1, MIA PaCa-2, BxPC-3, and ASPC-1 cancer cell lines, the IC_50_ values after 72 h of treatment were 18 ± 0.4 µM, 26 ± 1.4 µM, 14 ± 1.4 µM, and 27 ± 0.8 µM, respectively [47]. Other studies also indicated that dragmacidins G (**70**) and H (**71**) (Figure 4) showed moderate cytotoxicity against HeLa cells, with IC_50_ values of 4.2 and 4.6 µM, respectively [16].

Synthetic pyrazine bisindole compounds **72** and **73** (Figure 4) were tested, and although compound **72** showed weak cytotoxicity, its N-methylated derivative **73** exhibited significant inhibitory activity against leukemia, non-small cell lung cancer, colon cancer, CNS cancer, ovarian cancer, kidney cancer, and breast cancer cell lines, with GI_50_ values ranging from 0.058 to 7.19 µM [43].

New pyrimidine bisindole and pyrazine bisindole compounds were synthesized as potential anticancer drugs. When screened across 60 in vitro human tumor cell lines (encompassing leukemia, non-small cell lung cancer, colon cancer, CNS cancer, ovarian cancer, kidney cancer, and breast cancer cell lines), compounds **74**, **76**, **77**, and **78** (Figure 4) exhibited notable cytotoxicity in the low micromolar range, with GI_50_ values less than 11.7 µM. The cytotoxicity of compound **74** was the most pronounced, with GI_50_ values ranging from 0.16 to 1.42 µM. Compound **75** showed selective inhibitory effects against the IGROV1 cell line [48].

## 6. Fascaplysin Family Bisindole Alkaloids and Their Derivatives

The researchers isolated fascaplysin (**79**) (Figure 5) from the Fijian sponge Fascaplysinopsis *Bergquist* sp. [25]. Fascaplysin exhibits a broad range of anticancer activities against various cancer cell types, including NSCLC and SCLC (small cell lung cancer) cancer cells [49], glioma cells [50], A2780 and OVCAR3 human ovarian cancer cell lines [51], melanoma cells [52], and others. Fascaplysin exhibits anti-cancer effects in various types of cancer cells by inhibiting CDK4 (cyclin-dependent kinase 4)-mediated cell cycle progression. It helps to improve the anticancer effect of PI3K-AKT-targeted drugs, and when used in combination with AMPK (Adenosine 5′-monophosphate-activated protein kinase) inhibitors, it can also effectively inhibit tumor growth. Fascaplysin inhibits the expression of several folate and purine metabolism-related genes, such as MTR (5-methyltetrahydrofolate-homocysteine methyltransferase) and DHFR (dihydrofolate reductase), and makes cells sensitive to MTX (methotrexate)-induced apoptosis [53]. A research team has found that fascaplysin can induce HL-60 cell death through a synergistic effect between apoptosis and autophagy. The role of autophagy is activated by inhibiting the PI3K/Akt/mTOR signaling pathway, leading to increased expression of LC3-II, ATG7, and Beclin-1 [54]. Fascaplysin can trigger autophagy in endothelial cells by activating the ROS pathway [55]. Fascaplysin can inhibit the expression of survivin. In malignant melanoma, colorectal cancer, and lung cancer cells, fascaplysin can significantly increase TRAIL (TNF-related apoptosis-inducing ligand)-induced cancer cell death. These results suggest that fascaplysin may be a potential drug to overcome resistance to TRAIL-based cancer therapies [56].

The in vitro antiproliferative activity of the synthesized fascaplysin derivative **80** (Figure 5) demonstrated IC_50_ values of 290 nM and 320 nM against HeLa (cervical cancer) and THP-1 (acute monocytic leukemia) cell lines, respectively, while the IC_50_ values of fascaplysin were 550 nM and 890 nM, respectively. This indicates that the inhibitory activity of fascaplysin derivatives with phenyl substituents at the C-7 position on selected cancer cell lines is 2–3 times higher than that of fascaplysin itself [57].

Lyakhova’s team synthesized four fascaplysin derivatives, **80**–**83** (Figure 5). Cytotoxicity test results indicated that all compounds exhibited higher cytotoxicity than unsubstituted fascaplysin. Among these, 7-phenyl fascaplysin (**80**) and 3-bromo fascaplysin (**82**) were particularly effective against glioma C6 cells, with apoptosis being the primary mechanism of cell death. The cytotoxicity of these compounds was time- and concentration-dependent. Fascaplysin derivatives altered all stages of the glioma cell life cycle, and both significantly reduced the number of viable tumor cells in the G0 phase [58]. Compound **81** inhibits VEGF (vascular endothelial growth factor)-mediated angiogenesis, induces autophagy and apoptosis, and interferes with the PI3K/Akt/mTOR signaling pathway in MDAMB-231 cells [59].

## 7. Bisindole Acetylamine Alkaloids and Their Derivatives

2,2-bis (6-bromo-3-indolyl) ethylamine (**84**) (Figure 6), also known as BrBin, was isolated from the new caledonian sponge Orina and showed high cytotoxicity against U937, MCF-7, and Caco-2 cancer cell lines [60,61]. BrBin is a potent apoptotic agent. Its mechanism was first evaluated in the U937 tumor cell model. Studies revealed that BrBin modulates the ratio between anti-apoptotic and pro-apoptotic factors and induces mitochondria to participate in the activation of apoptosis mechanisms through the down-regulation of Bcl-2/Bcl-xL and the up-regulation of Bax, thus promoting apoptotic cell death in U937. The compound’s effect is dose-dependent [62].

Under similar conditions, compounds **85** and **86** (Figure 6) reduced U937 cell viability by about 10% compared to BrBin, while compound **87** (Figure 6) induced more cell death. Consequently, compound **87** maintained cytotoxicity equivalent to that observed after BrBin treatment. The structure-activity relationship studies indicated that the presence of a bromine atom and N-methylation are essential for apoptotic activity, and compound **87** functions in driving apoptotic cell death through mitochondrial involvement [63].

## 8. Acyclic Structure-Linked Bisindole Alkaloids and Their Derivatives

Calcicamide A (**88**) and B (**89**) (Figure 7) were isolated from the Irish sponge S. Calcicola. The cytotoxicity of calcicamide B against HeLa cells was measured after 6 and 24 h of incubation, revealing IC_50_ values of 165 ± 7 and 146 ± 13 µM, respectively, stronger than that of calcicamide A (IC_50_ > 200 µM) [64].

Spongocarbamides A and B (**90**–**91**) (Figure 7), which connect indole groups through linear urea-type bonds, exhibited weak inhibitory effects against A549 and K562 cancer cell lines, with IC_50_ > 90 µM [65].

Bisindoles **92** and **93** (Figure 7) were extracted from the formosan red alga, laurencia brongniartii. Compound **92** showed an inhibitory effect against HT-29 and P388 cancer cells, while compound **93** had a cytotoxic effect on P388 cells. Specific IC_50_ values are not provided in the literature [66].

Dendridine A (**94**) (Figure 7) exhibited weak cytotoxicity to mouse leukemia L1210 cells, with an IC_50_ of 32.5 μg/mL [67].

Chondriamides A (**95**) and B (**96**) (Figure 7), two cytotoxic bisindole amides, were isolated from the red alga chondria sp. Cytotoxicity tests indicated that chondriamide A had an IC_50_ of 0.5 μg/mL against KB cells and 5 μg/mL against LOVO (Human Colorectal Carcinoma Cells) cells, while the activity of chondriamide B on these two cancer cell types was <1 μg/mL and 10 μg/mL, respectively [68].

Eusynstyelamide B (**97**) (Figure 7), a natural product isolated from ascidians, was tested for toxicity on MDA-MB-231 cells, resulting in an IC_50_ of 5.0 µM. Further cell cycle analysis revealed significant stagnation in the G2/M phase [69]. Eusynstyelamides D (**98**) and E (**99**) (Figure 7), obtained by acid degradation of guanidine, showed weak cytotoxicity against the melanoma cell line A-2058, with IC_50_ values of 57 and 114.3 µM, respectively [70].

A series of bisindolyl triazinone analogues synthesized by Reddymasu Srenivasulu’s team were tested for in vitro activity. Compounds **100** and **101** (Figure 7) exhibited significant antiproliferative activity against human cervical cancer cell lines, with IC_50_ values of 4.6 and 1.3 µM, respectively, and showed no cytotoxic effects on HEK-293 cell lines (human embryonic kidney cells) [71].

## 9. Indolecarbazole Bisindole Alkaloids and Their Derivatives

STU (Staurosporine) (**102**) (Figure 8), isolated from the culture medium of Streptomyces marinus, is one of the most potent natural PKC (protein kinase C) inhibitors identified to date. It has shown considerable anti-tumor potential and has been extensively studied as a lead compound [72]. Staurosporine can sensitize cancer cells to cisplatin by down-regulating the p62 mechanism. In other cancer models, the combination of staurosporine and cisplatin may reduce cell proliferation and help overcome cisplatin resistance [73]. It can also induce apoptosis in pancreatic cancer cells (PaTu 8988t and Panc-1) through the intrinsic signaling pathway [74]. Staurosporine significantly induces autophagy by inhibiting the PI3K/Akt/mTOR signaling pathway, thereby inducing HepG2 cell death [75]. The ATM (ATM Serine/Threonine Kinase) inhibitor (AZD0156) and staurosporine inhibit the phosphorylation of TET1 (tet methylcysteine dioxygenase 1), causing instability of the TET1 protein and demonstrating a synergistic killing effect on B-ALL (B cell acute lymphoblastic leukemia) cells [76]. However, the poor target specificity and high toxicity of staurosporine have led to the failure of most preclinical studies.

Several staurosporine derivatives were isolated from the rice solid fermentation broth of the marine-derived *Streptomyces* sp. NB-A13. The inhibitory activity of compound **108** (Figure 8) on the SW-620 cell line was stronger than that of the positive control staurosporine, with an IC_50_ value of 9.99 nM, compared to staurosporine’s IC_50_ of 25.10 nM. Compounds **103**–**107**, **109**–**114**, and **116** (Figure 8) also exhibited notable cytotoxicity, with IC_50_ values ranging from 0.02 to 16.60 µM. Compound **108** (Figure 8) demonstrated strong cytotoxicity, suggesting that bridged sugar units are crucial for cytotoxicity. This was further evidenced by the increased cytotoxicity of compounds **107**, **111**, and **114** compared to other analogs. The PKC enzyme inhibition activity of compounds **103**–**108** was tested, with IC_50_ values ranging from 0.06 to 9.43 µM [77].

STU analogues generally exhibit lower toxicity compared to STU itself. Several of these analogues have been used in clinical and preclinical studies, including UCN-01 (**117**), lestaurtinib (**118**), and midostaurin (**119**) (Figure 8). UCN-01 is a selective inhibitor of protein kinase C [78] and has been shown to inhibit the protein expression of thymidylate synthase and E2F-1 (E2F transcription factor 1), as well as the DNA damage checkpoint kinase Chk1 (Check point kinase 1) [79]. It mediates Akt inactivation by inhibiting the upstream Akt kinase PDK1 (3-phosphoinositide-dependent protein kinase 1) [80] and can induce autophagy activation to protect cells from apoptosis, suggesting its potential as a standalone anticancer agent in treating human osteosarcoma [81]. UCN-01 inhibits cell proliferation in a group of NB (neuroblastoma) cell lines and detects the induction of cell apoptosis through caspase activation, caspase-3, and PARP cleavage [82]. AZD2461 (an effective PARP inhibitor)/UCN-01 combined inhibit PARP (poly (ADP ribose) polymer) and CHK1, reducing the expression of xbp1 (X-Box Binding Protein 1) in cells by downregulating c-Myc (cellular-myelocytomatosis viral oncogene) and upregulating the pro-apoptotic molecule CHOP (C/EBP-homologous protein**)**, leading to UPR (unfolded protein response) imbalance and cell death and trigger autophagy through PERK/eIF2alpha in MM (multiple myeloma) and IRE1alpha/JNK1/2 in PEL (primary effusion lymphoma) cells [83]. Lestaurtinib is a multi-target FLT3 (Fms-like Tyrosine Kinase-3) inhibitor (Figure 9) [84] with confirmed efficacy in inhibiting phosphorylation and inducing cell death in primary leukemia samples and in vivo mouse model studies [85]. In recent years, lestaurtinib has been shown to inhibit PKN1 (protein kinase N1), thereby affecting the SRF (serum response factor) activity induced by androgens in PCa (prostate cancer) cells and xenograft models. Many studies have shown that SRF plays a dominant role in the development and progression of various types of cancer [86]. The development history of lestaurtinib has been described in a review by Levis [87]. Midostaurin is the first oral multitargeted TKI (tyrosine kinase inhibitors) to improve overall survival in patients with FLT3-mutant AML (acute myeloid leukemia) and represents an important addition to the limited armamentarium against AML [88]. The first compound performed poorly in clinical trials due to its targeting effect. Lastly, lestaurtinib and midostaurin were developed for the treatment of acute myeloid leukemia with a positive FLT3 mutation (Figure 9) [89,90].

The study of UCN-01 and midostaurin has shown that modifications at the 3’-N and C-7 positions can improve the drug formation of staurosporine. Early studies on halogenated staurosporine derivatives revealed that halogens at the C-3 position could increase the selectivity and activity against some tumor cells [91,92,93]. This insight led to the synthesis of a series of staurosporine derivatives (**120**–**143**) (Figure 8) with modifications at the 3′-N, 3-, and 7- positions.

These compounds, except for **127**, exhibited strong inhibitory activity against the human acute myeloid leukemia MV4-11 cell line. For instance, compound **138** showed IC_50_ values of 0.078 µM and 0.666 µM against MV4-11 and PATU-8988T cells, respectively, with selection indexes of 1254 and 147. Compounds **134**, **137**, and **139** demonstrated IC_50_ values of 0.183 µM, 0.021 µM, and 0.029 µM against MCF-7 cells, respectively, with selection indexes of 76, 221, and 102. These values significantly surpassed the selectivity of the positive controls, UCN-01 (selection index 0.6) and doxorubicin (selection index 0.5). Additionally, compound **143** showed strong selective inhibitory activity against human colon cancer HCT-116 cells with an IC_50_ of 0.032 µM and a selection index of 19, which is more favorable than doxorubicin’s selection index of 0.04. Due to their high efficacy and low toxicity, these compounds hold potential as anticancer drugs [94]. Research has shown that compound **138** can regulate the ERK1/2, Akt, and p38 MAPK signaling pathways in MDA-MB-231 cells, as well as downregulate myc. In addition, in situ MDA-MB-231 xenotransplantation and the 4T1 syngeneic model showed that **138** has the effect of increasing tumor necrosis core and reducing lung and liver metastasis in mice [95].

Moreover, enzastaurin (**144**) (Figure 8 and Figure 9) was initially studied as an isoenzyme-specific derivative of staurosporine, targeting and inhibiting PKC β, an activator of tumor development. Currently, it is in clinical phase III trials but faces several challenges. The progress in the research and development of the anticancer drug enzastaurin has been detailed in a review by Johnston, R.N [96].

## 10. Other Bisindole Alkaloids

Spongosoritins A-D (**145**–**148**) (Figure 10), which connect two indole parts through a 2-methoxy-1-imidazole-5-one core, have been found to exhibit weak inhibitory effects against A549 and K562 cancer cell lines, with IC_50_ values ranging from 24.2 to 77.3 µM [65].

(±)-spondomine (**149**–**150**) (Figure 10), a cyclopentaindole alkaloid, has been isolated, with (+)-spondomine demonstrating strong cytotoxicity against the K562 tumor cell line with an IC_50_ of 2.2 µM. It is considered an attractive dual inhibitor of Wnt and HIF1. At 5 µM, the inhibitory activity of **149** was stronger than that of its positive control [97].

Hiroyasu Sato’s team reported that four new bisindole alkaloids with an imidazolinone ring, rhopaladins A-D (**151**–**154**) (Figure 10), were isolated from the Okinawan marine tunicate *Rhopalaea* sp. Rhopaladin B exhibited inhibitory activity against CDK4 (cyclin-dependent kinase 4) and c-erbB-2 kinase, with IC_50_ values of 12.5 and 7.4 μg/mL, respectively, indicating potential significant cytotoxicity to human tumor cells [98].

Hyrtinadine A (**155**) (Figure 10) has shown strong cytotoxicity against mouse leukemia L1210 and human epidermoid carcinoma KB cell lines, with IC_50_ values of 1 and 3 mg/mL, respectively, and weak cytotoxicity against HCT116 and A2780 cancer cells (IC_50_ > 10 µM). Interestingly, its precursor compound **156** (Figure 10) exhibited stronger activity, with IC_50_ values of 3.7 and 4.5 µM against HCT116 and A2780 cancer cells, respectively [99,100].

New bisindole alkaloids with a [5] spiro ring, spiroindimicins B-D (**157**–**159**) (Figure 10), were isolated from a deep-sea streptomyces strain identified by PCR (Polymerase Chain Reaction) screening. These compounds showed moderate cytotoxicity against several cancer cell lines. Spiroindimicin B had moderate cytotoxicity against CCRF-CEM, B16, and H460 cells. Spiroindimicin C inhibited the growth of HepG2 and H460 cells, while spiroindimicin D exhibited moderate inhibition on HepG2, B16, and H460 [101].

Indimicin B (**160**) (Figure 10), a bisindole alkaloid isolated from the marine streptomyces SCSIO 03032, features a unique 1′, 3′-dimethyl-2′-hydroindole moiety. It has demonstrated moderate cytotoxicity against the MCF-7 cell line, although specific activity data are not mentioned in the literature [102].

Luteoalbusins A and B (**161**–**162**), T988A (**163**), and gliocladines C and D (**164**–**165**) (Figure 10) were extracted from deep-sea fungi. These compounds exhibited significant cytotoxicity against SF-268 and MCF-7 cancer cells, with IC_50_ values ranging from 0.23 to 2.39 μM. Additionally, luteoalbusins A and B have anti-proliferative effects on NCI-H460 cells, with IC_50_ values of 1.15 to 1.31 μM [103].

Over the years, there has been extensive exploration of using telomerase inhibitors to inhibit cancer cell activity. Research indicates that telomerase plays a role in the carcinogenesis process in most cells. Telomerase activity continuously extends telomere length, leading to the uncontrolled division of cancer cells. Hence, overexpression of telomerase activity can result in genomic instability and replication immortality in 80–90% of malignant tumors [104]. Dictyodendrins A and B (**166**–**167**) (Figure 10), a class of alkaloids isolated from the sponge Dictyodendrlla verongiformis in southern Japan, have shown that the sulfate part of the molecule is essential for activity, as desulfurization compounds are completely inactive. They exhibit complete inhibition of telomerase at 50 μg/mL, marking them as the first marine natural products found to have inhibitory activity on telomerase [105].

## 11. Conclusions

The ocean serves as a rich source of compounds for drug development, with many marine extracts exhibiting varying degrees of anti-cancer activity. These compounds are pivotal in the discovery and development of anti-cancer drugs. Over the past decades, several marine bisindole alkaloids with anti-cancer properties have been identified, including the topsentins, nortopsentins, hamacanthins, dragmacidins, bisindolyl ethylamines, and fascaplysins families. Their novel molecular structures and broad biological and pharmacological activities have generated significant interest. Moreover, these alkaloids have been further modified to synthesize additional compounds with anti-cancer properties. This paper reviews 167 types of marine bisindole alkaloids and their derivatives (Table 1). In the future, marine natural products will become a focal point in drug discovery and development. These compounds, or elements of them, may provide foundational structures for new anti-tumor drug development. An increasing number of marine natural products are expected to be utilized in clinical treatments.

## Figures and Tables

**Figure 1 molecules-29-00933-f001:**
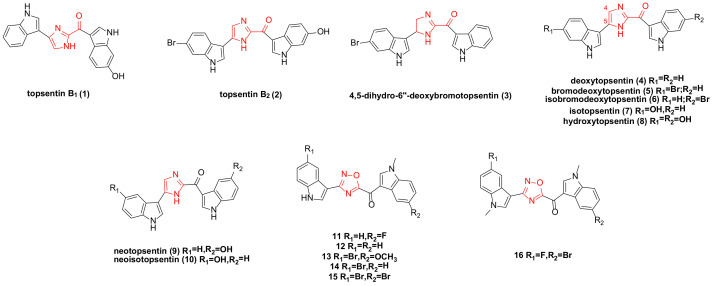
Topsentin family bisindole alkaloids and their derivatives.

**Figure 2 molecules-29-00933-f002:**
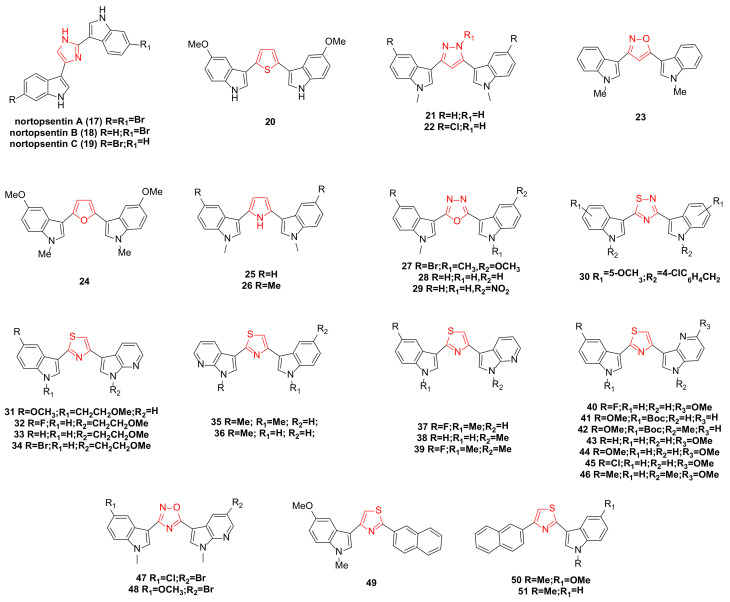
Noropsentin family bisindole alkaloids and their derivatives.

**Figure 3 molecules-29-00933-f003:**
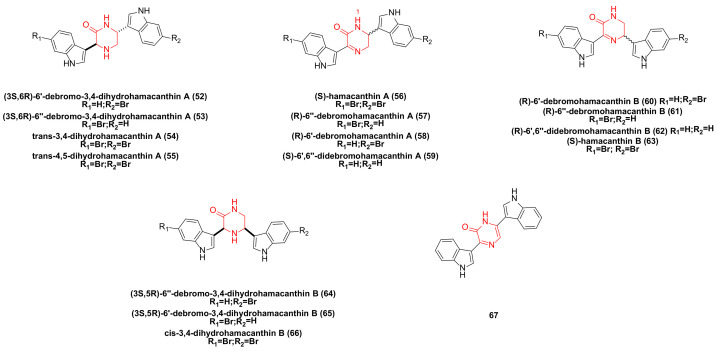
Hamacanthin family bisindole alkaloids and their derivatives.

**Figure 4 molecules-29-00933-f004:**
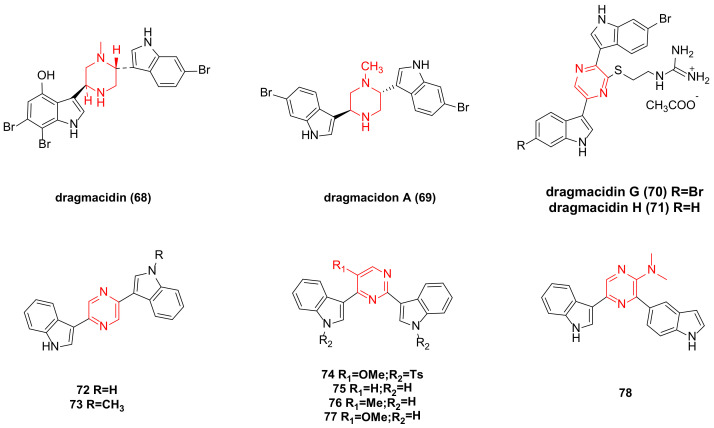
Dragmacidin family bisindole alkaloids and their derivatives.

**Figure 5 molecules-29-00933-f005:**
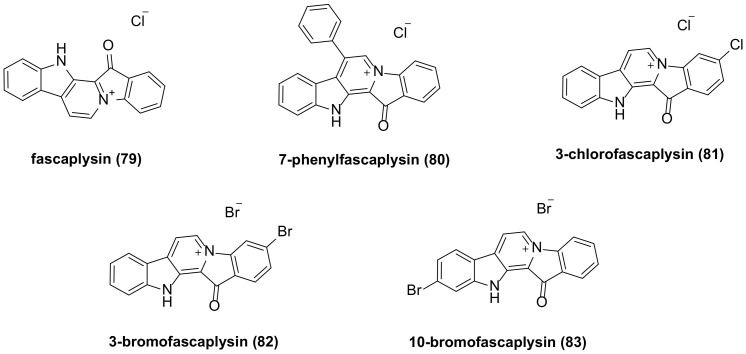
Fascaplysin family bisindole alkaloids and their derivatives.

**Figure 6 molecules-29-00933-f006:**
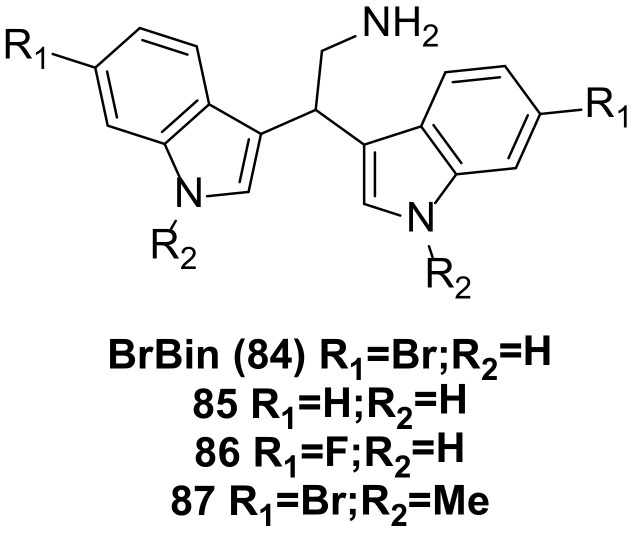
Bisindole acetylamine alkaloids and their derivatives.

**Figure 7 molecules-29-00933-f007:**
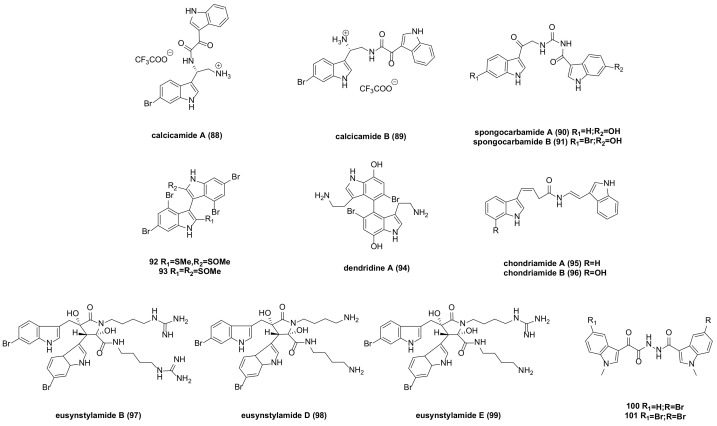
Acyclic structure linked bisindole alkaloids and derivatives.

**Figure 8 molecules-29-00933-f008:**
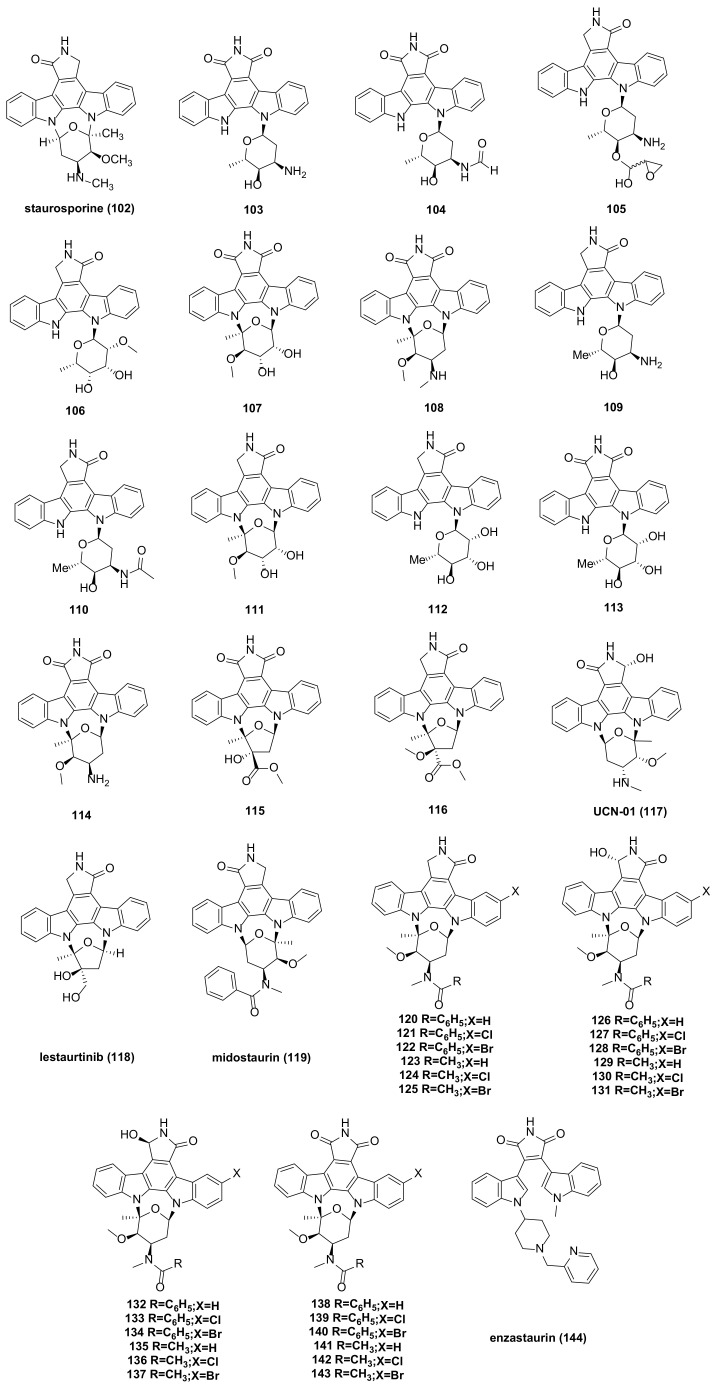
Indolecarbazole bisindole alkaloids and their derivatives.

**Figure 9 molecules-29-00933-f009:**
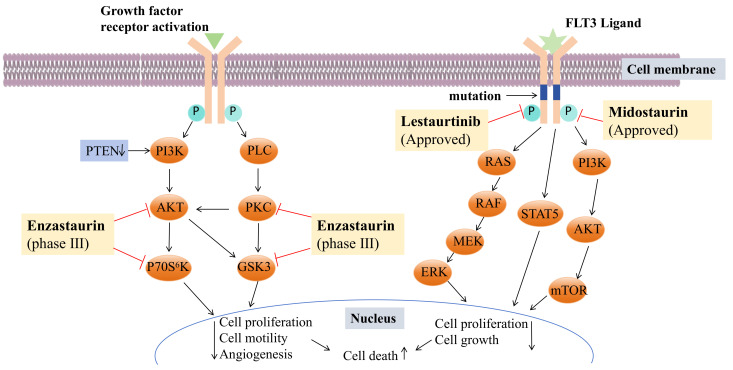
Mechanisms of enzastaurin, midostaurin, and lestaurtinib. (FLT3: Fms-like TyrosineKinase-3, PKC: protein kinase C, PTEN: phosphatase and tensin homolog deleted on chromosome ten, PLC: phospholipase C, PI3K: phosphoinositide-3 kinase, AKT also known as PKB: protein kinase B, GSK3: glycogen synthase kinase 3, RAS/RAF/MEK/ERK: mitogen-activated protein kinase signaling pathway, STAT5: Signal transducer and activator of transcription 5, mTOR: mammalian target of rapamycin).

**Figure 10 molecules-29-00933-f010:**
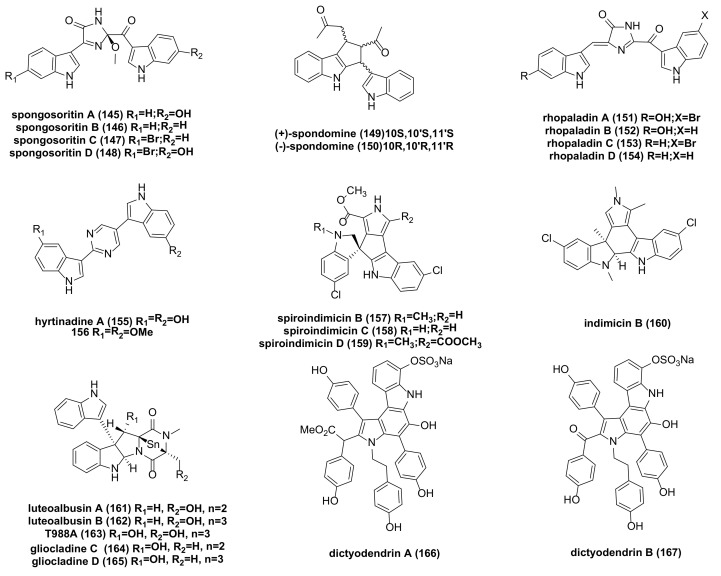
Other bisindole alkaloids.

**Table 1 molecules-29-00933-t001:** Anticancer activity of marine bisindole natural products and their derivatives.

Compound	Anticancer Activity	CAS Number	References
topsentin B_1_ (**1**)	IC_50_ (P388) = 4.1 ± 1.4 µM; IC_50_ (HT-29) = 20.5 ± 2.1 µM; IC_50_ (A549) = 41.1 ± 8.5 µM; IC_50_ (HL-60) = 15.7 ± 4.3 µM; IC_50_ (HCT-8;T47D) = 20 µg/mL; IC_50_ (NSCLC-N6) = 12 µg/mL; IC_50_ (HeLa) = 4.4 µM.	112515-44-2	[12,13,14,15,16]
topsentin B_2_ (**2**)/bromotopsentin	IC_50_ (NSCLC-N6) = 6.3 µg/mL; IC_50_ (HeLa) = 1.7 µM; IC_50_ (AGS) = 1.4 µg/mL; IC_50_ (L1210) = 6.9 µg/mL; IC_50_ (P388) = 7.0 µg/mL.	112515-44-3	[12,14,15,16,17,18]
4,5-dihydro-6″-deoxybromotopsentin (**3**)	IC_50_ (AGS) = 6.3 µg/mL; IC_50_ (L1210) = 5.1 µg/mL; IC_50_(BC) = 17.5 µg/mL; IC_50_ (P388) = 4.0 µg/mL.	116747-40-1	[12,17]
deoxytopsentin (**4**)	IC_50_ (AGS) = 1.3 µg/mL; IC_50_ (L1210) = 7.4 µg/mL; IC_50_ (BC) = 10.7 µg/mL; IC_50_ (HepG2) = 3.3 µg/mL; IC_50_ (P388) = 12.0 µg/mL.	112515-42-1	[12,17]
bromodeoxytopsentin (**5**)	IC_50_ (AGS) = 3.3 µg/mL; IC_50_ (L1210) = 1.1 µg/mL; IC_50_ (K-562) = 0.6 µg/mL.	180633-55-0	[17,19]
isobromodeoxytopsentin (**6**)	IC_50_ (K-562) = 2.1 µg/mL; ED_50_ (A549) = 12.3 µg/mL; ED_50_ (SK-OV-3) = 8.7 µg/mL; ED_50_ (SK-MEL-2) = 4.54 µg/mL; ED_50_ (XF498) = 5.51 µg/mL; ED_50_ (HCT15) = 6.38 µg/mL.	223596-72-3	[19,20]
isotopsentin (**7**)	IC_50_ (P388) = 4.0 µg/mL.	116725-88-3	[12]
hydroxytopsentin (**8**)	IC_50_ (P388) = 0.3 µg/mL.	116725-89-4	
neotopsentin (**9**)	IC_50_ (P388) = 2.5 µg/mL.	116725-90-7	
neoisotopsentin (**10**)	IC_50_ (P388) = 1.8 µg/mL.	116725-91-8	
**11**	EC_50_ (Panc-1) = 0.8 µM; EC_50_ (Capan-1) = 1.2 µM; EC_50_ (SUIT-2) = 0.4 µM.	2691060-39-4	[26]
**12**	EC_50_ (Panc-1) = 1.6 µM; EC_50_ (Capan-1) = 1.3 µM; EC_50_ (SUIT-2) = 3.2 µM.	2691060-40-7	
**13**	EC_50_ (Panc-1) = 2.8 µM; EC_50_ (Capan-1) = 2.8 µM; EC_50_ (SUIT-2) = 7.1 µM.	2691060-45-2	
**14**	EC_50_ (Panc-1) = 6.8 µM; EC_50_ (Capan-1) = 2.6 µM; EC_50_ (SUIT-2) = 5.9 µM.	2691060-46-3	
**15**	EC_50_ (Panc-1) = 1.5 µM; EC_50_ (Capan-1) = 1.4 µM; EC_50_ (SUIT-2) = 1.9 µM.	2691060-48-5	
**16**	IC_50_ (Hs766T) = 5.7 ± 0.60 µM; IC_50_ (HPAF-II) = 6.9 ± 0.25 µM; IC_50_ (PDAC3) = 9.8 ± 0.70 µM; IC_50_ (PDAC3) = 10.7 ± 0.16 µM.	2986653-16-9	[27]
nortopsentins A (**17**)	IC_50_ (P388) = 7.6 µg/mL.	134029-43-9	[18]
nortopsentins B (**18**)	IC_50_ (P388) = 7.8 µg/mL.	134029-44-0	
nortopsentins C (**19**)	IC_50_ (P388) = 1.7 µg/mL.	134029-45-1	
**20**	GI_50_ (CCRF-CEM) = 0.34 µM; GI_50_ (HL-60) = 2.27 µM; GI_50_ (K-562) = 3.54 µM; GI_50_ (RPMI-8226) = 2.83 µM; GI_50_ (MOLT-4) = 1.91 µM; GI_50_ (NCI-H522) = 1.31 µM; GI_50_ (HT29) = 2.79 µM; GI_50_ (LOX IMVI) = 2.55 µM.	937803-70-8	[29]
**21**	GI_50_ (MOLT-4) = 4.75 µM; GI_50_ (SR) = 3.46 µM; GI_50_ (HOP-92) = 2.06 µM; GI_50_ (NCI-H460) = 4.48 µM; GI_50_ (HCC-2998) = 4.74 µM; GI_50_ (LOX IMVI) = 1.70 µM; GI_50_ (MCF-7) = 3.95 µM.	959682-88-3	[30]
**22**	GI_50_ (MOLT-4) = 1.55 µM; GI_50_ (SR) = 2.36 µM; GI_50_ (HOP-92) = 1.86 µM; GI_50_ (NCI-H460) = 2.35 µM; GI_50_ (HCC-2998) = 1.71 µM; GI_50_ (SF-539) = 1.81 µM; GI_50_ (LOX IMVI) = 1.70 µM; GI_50_ (SK-MEL-5) = 1.79 µM; GI_50_ (UACC-62) = 1.63 µM; GI_50_ (IGROV1) = 2.55 µM; GI_50_ (CAKI-1) = 1.70 µM; GI_50_ (MCF-7) = 2.64 µM; GI_50_ (BT-549) = 2.03 µM.	959682-91-8	
**23**	IC_50_ (A549) = 5.1 µg/mL; IC_50_ (LXFA 629L) = 4.2 µg/mL; IC_50_ (UXF 1138L) = 4.8 µg/mL	1237096-77-3	[31]
**24**	GI_50_ (HL-60) = 1.98 µM; GI_50_ (K-562) = 2.86 µM; GI_50_ (RPMI-8226) = 2.25 µM; GI_50_ (SR) = 1.63 µM; GI_50_ (SK-MEL-2) = 2.13 µM; GI_50_ (OVCAR-4) = 2.06 µM; GI_50_ (OVCAR-5) = 2.79 µM; GI_50_ (NCI/ADR-RES) = 2.37 µM; GI_50_ (T-47D) = 2.72 µM; GI_50_ (SF-539) = 2.47 µM; GI_50_ (KM12) = 2.73 µM.	1237096-74-0	
**25**	IC_50_ (BXF 1218L) = 0.72 µM; IC_50_ (BXF 1352L) = 0.68 µM; IC_50_ (LXFL 1121L) = 0.67 µM; IC_50_ (MEXF 1341L) = 0.52 µM; IC_50_ (MEXF 276L) = 0.22 µM; IC_50_ (PAXF PANC1) = 0.74 µM; IC_50_ (SXF SAOS2) = 0.72 µM; IC_50_ (UXF 1138L) = 0.72 µM.	2987346-14-3	[32]
**26**	IC_50_ (BXF 1218L) = 0.32 µM; IC_50_ (MEXF 1341L) = 0.19 µM; IC_50_ (MEXF 276L) = 0.11 µM; IC_50_ (PRXF PC3M) = 0.32 µM; IC_50_ (SXF SAOS2) = 0.33 µM.	2986741-15-3	
**27**	IC_50_ (MCF-7) = 1.8 ± 0.9 µM; IC_50_ (Hela) = 9.23 ± 0.58 µM.	2412145-35-6	[33]
**28**	IC_50_ (Hela) = 9.4 ± 0.37 µM.	2412145-36-7	
**29**	IC_50_ (MCF-7) = 2.6 ± 0.89 µM; IC_50_ (Hela) = 6.34 ± 0.56 µM; IC_50_ (A549) = 3.3 ± 0.85 µM.	2412145-38-9	
**30**	IC_50_ (LNCAP) = 14.6 µM.	1338059-36-1	[34]
**31**	GI_50_ (MDA-MB-468) = 0.03 µM; GI_50_ (T-47D) = 0.04 µM; GI_50_ (MCF-7) = 0.05 µM.	2113670-47-4	[35]
**32**	GI_50_ (KM12) = 1.77 µM; GI_50_ (MDA-MB-435) = 0.90 µM; GI_50_ (T-47D) = 0.65 µM.	2113670-59-8	
**33**	GI_50_ (KM12) = 0.76 µM; GI_50_ (MDA-MB-435) = 0.86 µM; GI_50_ (T-47D) = 0.42 µM.	2113670-62-3	
**34**	GI_50_ (MCF-7) = 1.79 µM; GI_50_ (T-47D) = 1.97 µM; GI_50_ (EKVX) = 2.05 µM.	2113670-63-4	
**35**	GI_50_ (MDA-MB-435) = 0.03 µM; GI_50_ (SR) = 0.06 µM; GI_50_ (NCI-H522) = 0.04 µM.	1873270-61-1	[36]
**36**	GI_50_ (MDA-MB-435) = 0.04 µM; GI_50_ (SR) = 0.14 µM; GI_50_ (NCI-H522) = 0.05 µM.	1900737-01-0	
**37**	IC_50_ (STO) = 0.49 ± 0.07 µM; IC_50_ (MesoII) = 25.12 ± 3.06 µM.	1450997-15-5	[37]
**38**	IC_50_ (STO) = 0.33 ± 0.07 µM; IC_50_ (MesoII) = 4.11 ± 0.22 µM.	1450997-40-6	
**39**	IC_50_ (STO) = 0.43 ± 0.11 µM; IC_50_ (MesoII) = 4.85 ± 0.64 µM.	1450997-21-3	
**40**	IC_50_ (MDA-MB-231) = 9.5 ± 3.3 µM; IC_50_ (MIAPACA-2) = 11.0 ± 2.1 µM; IC_50_ (PC3) = 9.3 ± 2.0 µM; IC_50_ (STO) = 8.1 ± 1.3 µM.	1639456-87-3	[38]
**41**	IC_50_ (MDA-MB-231) = 17.5 ± 1.8 µM; IC_50_ (MIAPACA-2) = 14.5 ± 1.3 µM; IC_50_ (PC3) = 15.5 ± 1.1 µM; IC_50_ (STO) = 13.4 ± 1.9 µM.	1639457-02-5	[38]
**42**	IC_50_ (MDA-MB-231) = 19.8 ± 2.4 µM; IC_50_ (MIAPACA-2) = 16.4 ± 1.5 µM; IC_50_ (PC3) = 17.2 ± 2.5 µM; IC_50_ (STO) = 16.0 ± 2.4 µM.	1639457-04-7	
**43**	IC_50_ (MDA-MB-231) = 15.6 ± 3.2 µM; IC_50_ (MIAPACA-2) = 14.5 ± 2.9 µM; IC_50_ (PC3) = 19.4 ± 3.2 µM; IC_50_ (STO) = 12.5 ± 2.6 µM.	1639457-05-8	
**44**	IC_50_ (MDA-MB-231) = 25.0 ± 3.7 µM; IC_50_ (MIAPACA-2) = 19.4 ± 1.9 µM; IC_50_ (PC3) = 5.4 ± 1.4 µM; IC_50_ (STO) = 7.0 ± 2.1 µM.	1639457-07-0	
**45**	IC_50_ (MDA-MB-231) = 7.2 ± 0.4 µM; IC_50_ (MIAPACA-2) = 9.1 ± 3.2 µM; IC_50_ (PC3) =8.6 ± 4.0 µM; IC_50_ (STO) = 4.1 ± 1.0µM.	1639457-08-1	
**46**	IC_50_ (MDA-MB-231) = 14.6 ± 1.7 µM; IC_50_ (MIAPACA-2) = 19.7 ± 2.5 µM; IC_50_ (PC3) = 19.6 ± 2.4 µM; IC_50_ (STO) = 13.0 ± 1.8 µM.	1639457-11-6	
**47**	IC_50_ (MCF-7) = 0.65 ± 0.05 µM; IC_50_ (HCT-116) = 1.93 ± 0.06 µM; IC_50_ (Hela) = 10.56 ± 0.98 µM.	2354325-51-0	[39]
**48**	IC_50_ (MCF-7) = 2.41 ± 0.23 µM; IC_50_ (HCT-116) = 3.55 ± 0.1 µM; IC_50_ (Hela) = 13.96 ± 1.41 µM.	2354325-54-3	
**49**	IC_50_ (MCF-7) = 2.13 ± 0.12 µM.	2173313-53-4	[40]
**50**	IC_50_ (MCF-7) = 3.26 ± 0.19 µM.	2173313-72-7	
**51**	IC_50_ (MCF-7) = 5.14 ± 0.34 µM.	2173313-75-0	
(3S,6R)-6′-debromo-3,4-dihydrohamacanthin A (**52**)	ED_50_ (A549) = 7.50 µg/mL; ED_50_ (SK-OV-3) = 12.10 µg/mL; ED_50_ (SK-MEL-2) = 13.10 µg/mL; ED_50_ (XF498) = 19.10 µg/mL; ED_50_ (HCT15) = 6.30 µg/mL.	264624-44-4	[41]
(3S,6R)-6′′-debromo-3,4-dihydrohamacanthin A (**53**)	ED_50_ (SK-OV-3) = 4.92 µg/mL.	264624-45-5	
trans-3,4-dihydrohamacnathin A (**54**)	ED_50_ (A549) = 8.28 µg/mL; ED_50_ (SK-OV-3) = 8.03 µg/mL; ED_50_ (SK-MEL-2) = 9.14 µg/mL; ED_50_ (XF498) = 6.88 µg/mL; ED_50_ (HCT15) = 5.35 µg/mL.	264624-43-3	[20]
trans-4,5-dihydrohamacanthin A (**55**)	IC_50_ (AGS) = 6.3 µg/mL; IC_50_ (L1210) = 5.3 µg/mL.	453509-60-9	[17]
hamacanthin A (**56**)	IC_50_ (AGS) = 3.9 µg/mL; IC_50_ (L1210) = 3.0 µg/mL; ED_50_ (A549) = 4.49 µg/mL; ED_50_ (SK-OV-3) = 5.24 µg/mL; ED_50_ (SK-MEL-2) = 5.44 µg/mL; ED_50_ (XF498) = 5.60 µg/mL; ED_50_ (HCT15) = 4.66 µg/mL.	160098-92-0	[17,41]
(R)-6′′-debromohamacanthin A (**57**)	ED_50_ (A549) = 5.61 µg/mL; ED_50_ (SK-OV-3) = 4.20 µg/mL; ED_50_ (SK-MEL-2) = 4.73 µg/mL; ED_50_ (XF498) = 4.12 µg/mL; ED_50_ (HCT15) = 3.58 µg/mL.	853998-18-2	[17,20]
(R)-6′-debromohamacanthin A (**58**)	IC_50_ (AGS) = 7.5 µg/mL; IC_50_ (L1210) = 9.0 µg/mL; ED_50_ (HCT15) = 26.91 µg/mL.	853998-19-3	[17,20]
(S)-6′,6′′-didebromohamacanthin A (**59**)	ED_50_ (A549) = 8.30 µg/mL; ED_50_ (SK-OV-3) = 11.50 µg/mL; ED_50_ (SK-MEL-2) = 5.00 µg/mL; ED_50_ (XF498) = 17.10 µg/mL; ED_50_ (HCT15) = 4.10 µg/mL.	925457-99-4	[41]
(R)-6′-debromohamacanthin B (**60**)	ED_50_ (A549) = 3.71 µg/mL; ED_50_ (SK-OV-3) = 8.50 µg/mL; ED_50_ (SK-MEL-2) = 7.60 µg/mL; ED_50_ (XF498) = 8.30 µg/mL; ED_50_ (HCT15) = 4.20 µg/mL.	869964-40-9	
6″-debromohamacanthin B (**61**)	IC_50_ (AGS) = 7.5 µg/mL; IC_50_ (L1210) = 7.7 µg/mL; ED_50_ (A549) = 7.86 µg/mL; ED_50_ (SK-OV-3) = 7.85 µg/mL; ED_50_ (SK-MEL-2) = 7.71 µg/mL; ED_50_ (XF498) = 9.21 µg/mL; ED_50_ (HCT15) = 6.31 µg/mL.	925458-01-1	[17,41]
(R)-6′,6″-didebromohamacanthin B (**62**)	ED_50_ (A549) = 11.70 µg/mL; ED_50_ (SK-OV-3) = 12.60 µg/mL; ED_50_ (SK-MEL-2) = 13.70 µg/mL; ED_50_ (XF498) = 24.10 µg/mL; ED_50_ (HCT15) = 4.79 µg/mL.	925458-00-0	[41]
hamacanthin B(**63**)	IC_50_ (AGS) = 5.1 µg/mL; IC_50_ (L1210) = 6.7 µg/mL; ED_50_ (A549) = 2.14 µg/mL; ED_50_ (SK-OV-3) = 2.61 µg/mL; ED_50_ (SK-MEL-2) = 1.59 µg/mL; ED_50_ (XF498) = 2.93 µg/mL; ED_50_ (HCT15) = 1.52 µg/mL.	160098-93-1	[17,41]
(3S,5R)-6″- debromo-3,4-dihydrohamacanthin B (**64**)	ED_50_ (A549) = 4.20 µg/mL; ED_50_ (SK-OV-3) = 6.00 µg/mL; ED_50_ (SK-MEL-2) = 7.10 µg/mL; ED_50_ (XF498) = 6.80 µg/mL; ED_50_ (HCT15) = 6.30 µg/mL.	264624-40-0	[41]
(3S,5R)-6′-debromo-3,4-dihydrohamacanthin B (**65**)	ED_50_ (A549) = 9.67 µg/mL; ED_50_ (SK-OV-3) = 5.67 µg/mL; ED_50_ (XF498) = 9.74 µg/mL.	264624-41-1	
cis-3,4-dihydrohamacanthin B (**66**)	ED_50_ (A549) = 3.41 µg/mL; ED_50_ (SK-OV-3) = 3.62 µg/mL; ED_50_ (SK-MEL-2) = 3.85 µg/mL; ED_50_ (XF498) = 3.22 µg/mL; ED_50_ (HCT15) = 2.83 µg/mL.	264624-39-7	[20]
**67**	GI_50_ (MCF-7) = 6.60 µM; GI_50_ (K-562) = 18.8 µM; GI_50_ (SW620) = 18.9 µM.	265111-00-0	[42]
dragmacidin (**68**)	IC_50_ (P388) = 15 µg/mL; 1-10 µg/mLagainst A-549, HCT-8 and MDAMB cancer cell lines.	10952015(PubChem CID)	[44]
dragmacidon A (**69**)	ED_50_ (L1210) = 10 mg/mL; IC_50_ (A549) = 3.1 µM; IC_50_ (HT29) = 3.3 µM; IC_50_ (MDA-MB-231) = 3.8 µM.	128364-31-8	[45,46]
dragmacidin G (**70**)	IC_50_ (PANC-1) = 18 ± 0.4 µM; IC_50_ (MIA PaCa-2) = 26 ± 1.4 µM; IC_50_ (BxPC-3) = 14 ± 1.4 µM; IC_50_ (ASPC-1) = 27 ± 0.8 µM; IC_50_ (HeLa) = 4.2 µM.	2044674-17-9	[16,47]
dragmacidin H(**71**)	IC_50_ (HeLa) = 4.6 µM.	2650064-06-3	[16]
**72**	GI_50_ (MDA-N) = 2.47 µM; GI_50_ (RXF-393) = 2.73 µM; GI_50_ (K562) = 2.97 µM.	265110-98-3	[43]
**73**	GI_50_ (KM12) = 0.058 µM; GI_50_ (HCT-15) = 0.248 µM; GI_50_ (SK-MEL-5) = 0.287 µM.	/	
**74**	GI_50_ (SF-539) = 0.16 µM; GI_50_ (SR) = 0.22 µM; GI_50_ (MDA-MB-435) = 0.22 µM.	289503-31-7	[48]
**75**	GI_50_ (IGROV1) < 0.01 µM.GI_50_ (CCRF-CEM) = 1.51 µM; GI_50_ (HOP-92) = 1.70 µM.	360062-41-5	[48]
**76**	GI_50_ (IGROV1) = 1.14 µM.GI_50_ (CCRF-CEM) = 1.52 µM; GI_50_ (SN12C) = 1.68 µM.	360062-43-7	
**77**	GI_50_ (CCRF-CEM) = 1.13 µM; GI_50_ (IGROV1) = 1.86 µM; GI_50_ (HOP-92) = 2.34 µM.	360062-45-9	
**78**	GI_50_ (SNB-19) = 1.15 µM; GI_50_ (NCI-H460) = 2.12 µM; GI_50_ (MCF-7) = 2.30 µM.	485830-58-8	
fascaplysin(**79**)	Fascplysin has a wide range of anti-cancer activities, including NSCLC and SCLC cancer cells, glioma cells, A2780 and OVCAR3 human ovarian cancer cell lines, melanoma cells, etc.	114719-57-2	[49,50,51,52,53,54,55,56]
7-phenyl fascaplysin (**80**)	IC_50_ (HeLa) = 290 nM; IC_50_ (THP-1) = 320 nM; have a significant killing effect against glioma C6 cells.	2176439-03-3	[57,58]
3-chloro fascaplysin (**81**), 3-bromo fascaplysin (**82**),10-bromo fascaplysin (**83**)	Have a significant killing effect against glioma C6 cells. No accurate data on activity are provided in the literature.	1827526-66-8;959930-44-0;959930-45-1	[58]
BrBin(**84**)	IC_50_ (MCF-7) = 4.4 ± 1.9 µM; IC_50_ (Caco-2) = 1.5 ± 0.2 µM; GI_50_ (U937) values in the 1-1.3 µM range.	135077-20-2	[60,61]
**85**,**86**,**87**	Compounds 87 and 88 reduced U937 cell viability by about 10% compared with BrBin, and compound 89 induced more dead cells. No accurate data on activity are provided in the literature.	3616-44-2;2144744-62-5;1905457-00-2	[63]
calcicamide A (**88**)	IC_50_(HeLa) >200 µM.	/	[64]
calcicamide B (**89**)	IC_50_ (HeLa) = 146 ± 13 µM.	/	
spongocarbamide A (**90**)	IC_50_ (Srt A) = 79.4 µM; IC_50_ (K562) = 92.8 µM.	/	[65]
spongocarbamide B (**91**)	IC_50_ (Srt A) = 52.4 µM	/	
**92**	Have inhibitory effect against HT-29 and P388 cancer cells. No accurate data on activity are provided in the literature.	854781-00-3	[66]
**93**	Have an inhibitory effect against P388 cells. No accurate data on activity are provided in the literature.	854781-01-4	
dendridine A (**94**)	IC_50_ (L1210) = 32.5 µg/mL.	862844-50-6	[67]
chondriamide A (**95**)	IC_50_ (KB) = 0.5 µg/mL; IC_50_ (LOVO) = 5 µg/mL.	142677-09-6	[68]
chondriamide B (**96**)	IC_50_ (KB) < 1 µg/mL; IC_50_ (LOVO) = 10 µg/mL.	142677-10-9	
eusynstyelamide B (**97**)	IC_50_ (MDA-MB-231) = 5.0 µM.	1166392-99-9	[69]
eusynstyelamide D (**98**)	IC_50_ (A-2058) = 57 µM.	1280219-08-0	[70]
eusynstyelamide E (**99**)	IC_50_ (A-2058) = 114.3 µM.	1280219-14-8	
**100**	Showed significant antiproliferative activity against human cervical cancer cell lines, with IC_50_ values of 4.6 µM.	2226102-19-6	[71]
**101**	Showed significant antiproliferative activity against human cervical cancer cell lines, with IC_50_ values of 1.3 µM.	2226102-19-6	
staurosporine (**102**)	It has good anti-tumor potential and has been widely studied as a lead compound. IC_50_ (PC-3) = 58.94 ± 2.30 nM; IC_50_ (SW-620) = 25.10 ± 3.20 nM	62996-74-1	[72,73,74,75,76,77]
**103**	IC_50_ (PC-3) = 4.03 ± 0.29 µM; IC_50_ (SW-620) = 2.14 ± 0.08 µM.	2242549-12-6	[77]
**104**	IC_50_ (PC-3) = 2.05 ± 0.06 µM; IC_50_ (SW-620) = 0.74 ± 0.01 µM.	2242549-13-7	
**105**	IC_50_ (PC-3) = 2.45 ± 0.08 µM; IC_50_ (SW-620) = 2.00 ± 0.19 µM.	2102303-78-4	
**106**	IC_50_ (PC-3) = 16.60 ± 0.43 µM; IC_50_ (SW-620) = 9.54 ± 0.65 µM.	2242549-14-8	
**107**	IC_50_ (PC-3) = 0.55 ± 0.04 µM; IC_50_ (SW-620) = 0.16 ± 0.01 µM.	2102303-76-2	
**108**	IC_50_ (PC-3) = 56.22 ± 0.54 nM; IC_50_ (SW-620) = 9.99 ± 0.86 nM.	125035-83-8	
**109**	IC_50_ (PC-3) = 2.06 ± 0.12 µM; IC_50_ (SW-620) = 0.76 ± 0.04 µM.	249512-77-4	
**110**	IC_50_ (PC-3) = 2.50 ± 0.03 µM; IC_50_ (SW-620) = 0.73 ± 0.03 µM.	2251748-53-3	
**111**	IC_50_ (PC-3) = 0.56 ± 0.04 µM; IC_50_ (SW-620) = 0.18 ± 0.00 µM.	155416-34-5	
**112**	IC_50_ (PC-3) = 1.87 ± 0.03 µM; IC_50_ (SW-620) = 1.04 ± 0.04 µM.	105114-22-5	
**113**	IC_50_ (PC-3) = 2.27 ± 0.06 µM; IC_50_ (SW-620) = 0.81 ± 0.03 µM.	1185908-87-5	
**114**	IC_50_ (PC-3) = 0.69 ± 0.02 µM; IC_50_ (SW-620) = 0.47 ± 0.02 µM.	160256-54-2	
**115**	IC_50_ (PC-3) >20 µM; IC_50_ (SW-620) = >20 µM.	156582018(PubChem CID)	
**116**	IC_50_ (PC-3) = 0.77 ± 0.05 µM; IC_50_ (SW-620) = 0.02 ± 0.00 µM.	187810-82-8	
UCN-01 (**117**)	Entered clinical trials, but with poor targeting.IC_50_ (MCF-7) = 1.636 ± 0.281 µM; IC_50_ (L-02) = 1.034 ± 0.111 µM;	112953-11-4	[78,79,80,81,82,83]
lestaurtinib (**118**)	Entered clinical trials, but with poor targeting.	111358-88-4	[84,85,86,87]
midostaurin (**119**)	Midostaurin was developed and listed as a drug on the market, mainly for the treatment of acute myeloid leukemia with a positive FLT3 mutation.	120685-11-2	[88,89,90]
**120**	IC_50_ (MV4-11) = 0.020 ± 0.010 µM; IC_50_ (MCF-7) = 0.235 ± 0.017 µM; IC_50_ (HCT-116) = 0.439 ±0.120 µM.	120685-11-2	[94]
**121**	IC_50_ (MV4-11) = 0.027 ± 0.000 µM; IC_50_ (MCF-7) = 0.122 ± 0.018 µM; IC_50_ (HCT-116) = 0.098 ±0.022 µM.	2771013-95-5	
**122**	IC_50_ (MV4-11) = 0.031 ± 0.000 µM; IC_50_ (MCF-7) = 0.257 ± 0.035 µM; IC_50_ (HCT-116) = 0.141 ±0.013 µM.	2771013-97-7	
**123**	IC_50_ (MV4-11) = 0.011 ± 0.000 µM; IC_50_ (MCF-7) = 0.121 ± 0.033 µM; IC_50_ (HCT-116) = 0.149 ±0.032 µM.	2173322-11-5	
**124**	IC_50_ (MV4-11) = 0.015 ± 0.001 µM; IC_50_ (MCF-7) = 0.018 ± 0.010 µM; IC_50_ (HCT-116) = 0.090 ±0.007 µM.	2771013-94-4	
**125**	IC_50_ (MV4-11) = 0.012 ± 0.001 µM; IC_50_ (MCF-7) = 0.072 ± 0.028 µM; IC_50_ (HCT-116) = 0.032 ±0.006 µM.	2771013-96-6	
**126**	IC_50_ (MV4-11) = 0.275 ± 0.009 µM; IC_50_ (L-02) = 43.530 ±0.750 µM.	945260-14-0	
**127**	IC_50_ (L-02) = 67.903 ±1.678 µM.	2771014-05-0	
**128**	IC_50_ (MV4-11) = 0.295 ± 0.005 µM; IC_50_ (L-02) = 59.807 ±0.248 µM.	2771014-09-4	
**129**	IC_50_ (MV4-11) = 0.128 ± 0.005 µM; IC_50_ (L-02) = 28.780 ±1.530 µM.	137888-66-5(CAS number of racemate)	
**130**	IC_50_ (MV4-11) = 0.135 ± 0.004 µM; IC_50_ (L-02) = 8.740 ± 0.513 µM.	2771014-03-8	
**131**	IC_50_ (MV4-11) = 0.139 ± 0.012 µM; IC_50_ (L-02) = 4.432 ± 0.785 µM.	2771014-07-2	
**132**	IC_50_ (MV4-11) = 0.238 ± 0.014 µM; IC_50_ (L-02) = 23.32 ± 1.237 µM.	155848-20-7	
**133**	IC_50_ (MV4-11) = 0.635 ± 0.020 µM; IC_50_ (L-02) = 12.325 ±0.191 µM.	2771014-04-9	
**134**	IC50 (MV4-11) = 0.328 ± 0.022 µM; IC_50_ (MCF-7) = 0.183 ± 0.032 µM; IC_50_ (L-02) = 13.875 ±0.248 µM.	2771014-08-3	
**135**	IC_50_ (MV4-11) = 0.148 ± 0.004 µM; IC_50_ (MCF-7) = 0.621 ± 0.185 µM; IC_50_ (L-02) = 18.490 ±0.433 µM.	137888-66-5(CAS number of racemate)	
**136**	IC_50_ (MV4-11) = 0.582 ± 0.072 µM; IC_50_ (L-02) = 8.635 ± 0.279 µM.	2771014-02-7	
**137**	IC_50_ (MV4-11) = 0.217 ± 0.016 µM; IC_50_ (MCF-7) = 0.021 ± 0.002 µM; IC_50_ (HCT-116) = 0.793 ±0.201 µM.	2771014-06-1	
**138**	IC_50_ (MV4-11) = 0.078 ±0.004 µM; IC_50_ (PATU-8988T) = 0.666 ± 0.055 µM; IC_50_ (L-02) = 97.800 ±0.297 µM.	154589-96-5	
**139**	IC_50_ (MV4-11) = 0.073 ± 0.006 µM; IC_50_ (MCF-7) = 0.029 ± 0.002 µM; IC_50_ (L-02) = 2.977 ± 0.405 µM.	2771013-99-9	
**140**	IC_50_ (MV4-11) = 0.061 ± 0.005 µM; IC_50_ (L-02) = 3.238 ± 0.030 µM.	2771014-01-6	
**141**	IC_50_ (MV4-11) = 0.091 ± 0.003 µM; IC_50_ (MCF-7) = 0.312 ± 0.045 µM; IC_50_ (HCT-116) = 0.245 ±0.007 µM.	125035-81-6	
**142**	IC_50_ (MV4-11) = 0.038 ± 0.006 µM; IC_50_ (MCF-7) = 0.014 ± 0.002 µM; IC_50_ (HCT-116) = 0.058 ±0.006 µM.	2771013-98-8	
**143**	IC_50_ (MV4-11) = 0.013 ± 0.001 µM; IC_50_ (MCF-7) = 0.016 ± 0.002 µM; IC_50_ (HCT-116) = 0.032 ±0.003 µM.	2771014-00-5	
spongosoritin A (**145**)	IC_50_ (A549) = 77.3 µM; IC_50_ (K562) = 24.2 µM.	/	[65]
spongosoritin B (**146**)	IC_50_ (A549) = 55.7 µM; IC_50_ (K562) = 28.5 µM.	/	
spongosoritin C (**147**)	IC_50_ (A549) = 61.2 µM; IC_50_ (K562) = 37.7 µM.	/	
spongosoritin D (**148**)	IC_50_ (A549) = 70.9 µM; IC_50_ (K562) = 54.2 µM.	/	
(+)-spondomine (**149**)	IC_50_ (K562) = 2.2 µM.	2687275-02-9	[97]
rhopaladin B (**152**)	Showed inhibitory activity on cyclin dependent kinase 4 and c-erbB-2 kinase, with IC_50_ of 12.5 and 7.4 μg/mL.	212069-49-3	[98]
hyrtinadine A (**155**)	IC_50_ (L1210) = 1 mg/mL; IC_50_ (KB) = 3 mg/mL; weak cytotoxicity to HCT116 and A2780 cancer cells (IC_50_ >10 µM).	925253-33-4	[99]
**156**	IC_50_ (HCT116) = 3.7 µM; IC_50_ (A2780) = 4.5 µM.	1333469-91-2	[100]
spiroindimicin B (**157**)	IC_50_ (B16) = 5 µg/mL; IC_50_ (H460) = 12 µg/mL; IC_50_ (CCRF-CEM) = 4 µg/mL.	1380717-82-7	[101]
spiroindimicin C (**158**)	IC_50_ (HepG2) = 6 µg/mL; IC_50_ (H460) = 15 µg/mL;	1380717-83-8	[101]
spiroindimicin D (**159**)	IC_50_ (HepG2) = 22 µg/mL; IC_50_ (B16) = 20 µg/mL; IC_50_ (H460) = 18 µg/mL.	1380717-83-8	
indimicin B (**160**)	Moderate cytotoxicity against MCF-7 cell line.	1620987-80-5	[102]
luteoalbusin A (**161**)	IC_50_ (SF-268) = 0.46 ± 0.05 µM; IC_50_ (MCF-7) = 0.23 ± 0.03 µM; IC_50_ (NCI-H460) = 1.15 ± 0.03 µM; IC_50_ (HepG2) = 0.91 ± 0.03 µM.	1414774-32-5	[103]
luteoalbusin B (**162**)	IC_50_ (SF-268) = 0.59 ± 0.03 µM; IC_50_ (MCF-7) = 0.25 ± 0.00 µM; IC_50_ (NCI-H460) = 1.31 ± 0.12 µM; IC_50_ (HepG2) = 1.29 ± 0.16 µM.	1414774-33-6	
T988A (**163**)	IC_50_ (SF-268) = 1.04 ± 0.03 µM; IC_50_ (MCF-7) = 0.91 ± 0.03 µM; IC_50_ (NCI-H460) = 5.60 ± 0.58 µM; IC_50_ (HepG2) = 3.52 ± 0.74 µM.	823802-55-7	
gliocladine C (**164**)	IC_50_ (SF-268) = 0.73 ± 0.05 µM; IC_50_ (MCF-7) = 0.23 ± 0.03 µM; IC_50_ (NCI-H460) = 6.57 ± 0.81 µM; IC_50_ (HepG2) = 0.53 ± 0.04 µM.	871335-07-8	
gliocladine D (**165**)	IC_50_ (SF-268) = 2.49 ± 0.07 µM; IC_50_ (MCF-7) = 0.65 ± 0.07 µM; IC_50_ (NCI-H460) = 17.78 ± 0.27 µM; IC_50_ (HepG2) = 2.03 ± 0.07 µM.	871335-08-9	

## Data Availability

Not applicable.

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
