# Peer review of "Marine-Derived Bisindoles for Potent Selective Cancer Drug Discovery and Development"

_molecules, 2024, doi:10.3390/molecules29050933_

Round 1

Reviewer 1 Report

Comments and Suggestions for Authors

1. Please provide a list of abbreviations, or give the abbreviation the first time it is mentioned.

2. Please provide the PubChem CID or CAS number for each Marine-Derived Bisindoles discussed in this manuscript.

3. Give a statement on how and using what the FIGURES were made/produced? provide statements for each legendary figures.

4. Provide an interesting mechanism picture that explains the mechanism of action of "Marine-Derived Bisindoles" for its potential as an anticancer; to improve readability.

5. There is no further explanation from "Table 1. Anticancer activity of marine bisindole Natural products and their derivatives" which should show specifically that each compounds was tested on what specific cancer cells?

it must be specified. The table has also not been mentioned in the main manuscript where it must be referred to.

Author Response

Dear reviewer:

We would like to thank you for giving us constructive suggestions which would help us improve the quality of the paper. Here we submit a new version of our manuscript with the title “Marine-Derived Bisindoles for Potent Selective Cancer Drug Discovery and Development” (Manuscript ID: molecules-2793725), which has been modified according to your suggestions. The main corrections in the paper and responds to your comments were as as following:

Question 1: Please provide a list of abbreviations, or give the abbreviation the first time it is mentioned

Response to question 1: Thanks for the suggestions from the scrupulous reviewer. we have added the abbreviation when the first time it is mentioned.

Question 2: Please provide the PubChem CID or CAS number for each Marine-Derived Bisindoles discussed in this manuscript. 

Response to question 2: Thanks for the suggestions from the scrupulous reviewer. We have provided CAS or PubChem CID numbers in Table 1 for most marine-derived bisindoles. However, the CAS/PubChem CID number of compound 75, calcicamide A, calcicamide B, spongocarbamide A, spongocarbamide B, spongosoritin A, spongosoritin B, spongosoritin C and spongosoritin D have not been found in databases(Including Reaxys, SciFinder and PubChem).

Question 3: Give a statement on how and using what the FIGURES were made/produced? provide statements for each legendary figures. 

Response to question 3: Thanks for the suggestions from the scrupulous reviewer. We have provided statements for each legendary figures.

Question 4: Provide an interesting mechanism picture that explains the mechanism of action of "Marine-Derived Bisindoles" for its potential as an anticancer; to improve readability.

Response to question 4: Thanks for the suggestions. We have added a mechanism picture, as shown in figure 8.

Question 5: There is no further explanation from "Table 1. Anticancer activity of marine bisindole Natural products and their derivatives" which should show specifically that each compounds was tested on what specific cancer cells? it must be specified. The table has also not been mentioned in the main manuscript where it must be referred to.

Response to question 5: Thanks for your suggestions. We have revised our manuscript as per your suggestions.

Yours Sincerely,

Shuanglin Qin

Cell phone: +86-13972820791

E-mail addresses: shuanglin@tju.edu.cn

Reviewer 2 Report

Comments and Suggestions for Authors

The review is well organized and readable. It only looks like the authors have taken a section of a published review article https://www.sciencedirect.com/science/article/pii/S022352342200650X#sec2, yes they reference it, but distinction of the two reviews should be clarified. 

I have also given suggestions as scribbles on the attached. 

Comments on the Quality of English Language

The authors use capitals in text which are not necessary. They need to review this on the entire document. 

Author Response

Dear reviewer:

We would like to thank you for giving us constructive suggestions which would help us improve the quality of the paper. Here we submit a new version of our manuscript with the title “Marine-Derived Bisindoles for Potent Selective Cancer Drug Discovery and Development” (Manuscript ID: molecules-2793725), which has been modified according to your suggestions. The main corrections in the paper and responds to your comments were as as following:

Question 1: The review is well organized and readable. It only looks like the authors have taken a section of a published review article https://www.sciencedirect.com/science/article/pii/S022352342200650X#sec2, yes they reference it, but distinction of the two reviews should be clarified.   

Response to question 1: Thanks for the suggestions. We wrote this review based on the higher success rate of marine-derived products in drug development. And In this manuscript, we emphasize the unique mechanism and potent selective antitumor activity of marine-derived bisindoles.

Question 2: I have also given suggestions as scribbles on the attached. peer-review-34328882.v1.pdf

Response to question 2: Thanks for the suggestions. We have revised the incorrect structure and conducted a self-examination to correct all the incorrect structures. After comparing multiple literatures, we found that topsentin, also known as topsentin B1 (1). Topsentin B2 (2) and bromotopsentin are the same compound but named differently in different articles. The structure of 4,5-dihydro-6 '- deoxybromotopsentin (3) should be as shown in Figure 1.

Question 3: The authors use capitals in text which are not necessary. They need to review this on the entire document.

Response to question 3: Thanks for your suggestions. We have checked the entire manuscript and removed unnecessary capitals.

Yours Sincerely,

Shuanglin Qin

Cell phone: +86-13972820791

E-mail addresses: shuanglin@tju.edu.cn

Reviewer 3 Report

Comments and Suggestions for Authors

The paper presents a review of bisindole alkaloids obtained from marine sources and their derivatives promising for development of anti-cancer treatment. The review is comprehensive and well written. However some corrections are needed for paper improvement.

1. The Introduction section is far too concise. It would benefit from more information on the drugs obtained from marine sources that are alredy used in clinical practice, especially those with anti-cancer effect. Arguments supplying the choice of indole as a working compound for anti-cancer drugs should be added.

2. The text in the Conclusions sections should rather be moved to the Introduction. Conclusions are supposed to summarize the information presented in the paper and mark future directions of the research.

3. Please indicate throughout the text references to the Figures where the  structures of the discussed compounds are depicted.

Comments on the Quality of English Language

The English language of the paper is fine, but I've noticed some misprints.

Author Response

Dear reviewer:

We would like to thank you for giving us constructive suggestions which would help us improve the quality of the paper. Here we submit a new version of our manuscript with the title “Marine-Derived Bisindoles for Potent Selective Cancer Drug Discovery and Development” (Manuscript ID: molecules-2793725), which has been modified according to your suggestions. The main corrections in the paper and responds to your comments were as as following:

Question 1: The Introduction section is far too concise. It would benefit from more information on the drugs obtained from marine sources that are alredy used in clinical practice, especially those with anti-cancer effect. Arguments supplying the choice of indole as a working compound for anti-cancer drugs should be added.   

Response to question 1: Thanks for your suggestions. We have revised the Introduction section as per your suggestions.

Question 2: The text in the Conclusions sections should rather be moved to the Introduction. Conclusions are supposed to summarize the information presented in the paper and mark future directions of the research.

Response to question 2: Thanks for your suggestions. We have revised the manuscript as per your suggestions.

Question 3: Please indicate throughout the text references to the Figures where the  structures of the discussed compounds are depicted. 

Response to question 3: Thanks for your suggestions. We have indicated throughout the text references to the figures where the structures of the discussed compounds are depicted.

Yours Sincerely,

Shuanglin Qin

Cell phone: +86-13972820791

E-mail addresses: shuanglin@tju.edu.cn

Reviewer 4 Report

Comments and Suggestions for Authors

I have reviewed review article entitled, "Marine-Derived Bisindoles for Novel Cancer Drug Discovery and Development". Review article is good but i have few queries to improve quality of this work. These include:

1) Incorporate significant potentials in abstract section?

2) Incorporate a short note about techniques involved in isolation of marine derived Bisindoles.

3) Number of figures too high it can be reduced to 10.

4) Add a paragraph detailing physicochemical properties of marine based bisnidoles?

5) Thoroughly review for grammatical errors.

Many Thanks

Comments on the Quality of English Language

NA

Author Response

Dear reviewer:

We would like to thank you for giving us constructive suggestions which would help us improve the quality of the paper. Here we submit a new version of our manuscript with the title “Marine-Derived Bisindoles for Potent Selective Cancer Drug Discovery and Development” (Manuscript ID: molecules-2793725), which has been modified according to your suggestions. The main corrections in the paper and responds to your comments were as as following:

Question 1: Incorporate significant potentials in abstract section?   

Response to question 1: Thanks for your suggestions. We have added relevant description in the abstract section.

Question 2: Incorporate a short note about techniques involved in isolation of marine derived Bisindoles.   

Response to question 2: Thanks for your suggestions. We have added the techniques involved in isolation of marine-derived bisindoles in the introduction section.

Question 3: Number of figures too high it can be reduced to 10.  

Response to question 3: Thanks for your suggestions. We have reduced the number of figures to 10.

Question 4: Add a paragraph detailing physicochemical properties of marine based bisnidoles? 

Response to question 4: Thanks for your suggestions. We have added the relevant description in the introduction section.

Question 5: Thoroughly review for grammatical errors.

Response to question 5: Thanks for your suggestions. This review was edited for proper English language, grammar, punctuation, spelling, and overall style by one or more of the highly qualified native English-speaking editors at ZiboYimore Translation CO. LTD. And, the certification has been shown as follows.

Yours Sincerely,

Shuanglin Qin

Cell phone: +86-13972820791

E-mail addresses: shuanglin@tju.edu.cn

Reviewer 5 Report

Comments and Suggestions for Authors

ِDear Authors

The current review article is well designed and written, but it should be kept in the mind that other new reviews in the same field have been published in 2022 and 2023 (attached file), so the novelty of the work is very low, so in order to increase the novelty of the work, so that it can be published in the very valuable journal of Molecules, the following suggestions are provided, if these changes are not applied completely and in detail, this article is not recommended for publication in this journal.

1- All marine and terrestrial plant and animal secondary metabolites, including alkaloids and polyphenols, have antioxidant and anticancer properties. But the secondery metabolite is valuable that has a more selective anti-cancer effect, so that it has a lethal effect on cancer cells and tissues, but has the least negative effect on the healthy tissues and cells. Therefore, it is suggested that the title of this article be changed as follows and the text of the article be changed accordingly:

"Marine-Derived Bisindoles for Potent Selective Cancer Drug Discovery  and Development"

Author should kept in the mind that in the text of the article, after mentioning the selective properties of various Derived Bisindoles, the main part of the discussion about these results become the  detailed discussion about the "Mechanism of their Selective Action", which is less mentioned in other review articles.

Author Response

Dear reviewer:

We would like to thank you for giving us constructive suggestions which would help us improve the quality of the paper. Here we submit a new version of our manuscript with the title “Marine-Derived Bisindoles for Potent Selective Cancer Drug Discovery and Development” (Manuscript ID: molecules-2793725), which has been modified according to your suggestions. The main corrections in the paper and responds to your comments were as as following:

To reviewer 5:

Question 1: The current review article is well designed and written, but it should be kept in the mind that other new reviews in the same field have been published in 2022 and 2023 (attached file), so the novelty of the work is very low, so in order to increase the novelty of the work, so that it can be published in the very valuable journal of Molecules, the following suggestions are provided, if these changes are not applied completely and in detail, this article is not recommended for publication in this journal.

1- All marine and terrestrial plant and animal secondary metabolites, including alkaloids and polyphenols, have antioxidant and anticancer properties. But the secondery metabolite is valuable that has a more selective anti-cancer effect, so that it has a lethal effect on cancer cells and tissues, but has the least negative effect on the healthy tissues and cells. Therefore, it is suggested that the title of this article be changed as follows and the text of the article be changed accordingly:

"Marine-Derived Bisindoles for Potent Selective Cancer Drug Discovery and Development"

Author should kept in the mind that in the text of the article, after mentioning the selective properties of various Derived Bisindoles, the main part of the discussion about these results become the  detailed discussion about the "Mechanism of their Selective Action", which is less mentioned in other review articles.

Response to question 1: Thanks for your suggestions. We have carefully studied the attachment you provided, and have carried on the careful comparison with our paper. Indeed, the selective properties are important features for various marine-derived bisindoles. Therefore, the title you provide should be more appropriate for our article. We have changed the title to "Marine Derived Bisindoles for Potential Selective Cancer Drug Discovery and Development".

Yours Sincerely,

Shuanglin Qin

Cell phone: +86-13972820791

E-mail addresses: shuanglin@tju.edu.cn

Round 2

Reviewer 5 Report

Comments and Suggestions for Authors

Dear authors

Unfortunately, according to the reviewer's comment, you have only changed the title of the manuscript and have not made the necessary changes in the text of the manuscript according to the title change.

Author Response

Dear reviewer:

We would like to thank you for giving us constructive suggestions which would help us improve the quality of the paper. Here we submit a new version of our manuscript with the title “Marine-Derived Bisindoles for Potent Selective Cancer Drug Discovery and Development” (Manuscript ID: molecules-2793725), which has been modified according to your suggestions. The main corrections in the paper and responds to your comments were as as following:

Question 1: Unfortunately, according to the reviewer's comment, you have only changed the title of the manuscript and have not made the necessary changes in the text of the manuscript according to the title change.

Response to question 1: Thanks for your suggestions. In order to correspond with the modification of the title, we have added the selectivity of corresponding compounds towards the targets and pathways, including dragmacidonA, fascaplysin, compound 81, staurosporine, NCN-01, lestaurtinib, midosturanin, compound 138, etc. Added the mechanism diagram of the main compounds (Figure 8). And corresponding references have been added.

Yours Sincerely,

Shuanglin Qin

Cell phone: +86-13972820791

E-mail addresses: shuanglin@tju.edu.cn
